# A conserved neuropeptide system links head and body motor circuits to enable adaptive behavior

Shankar Ramachandran[1†], Navonil Banerjee[1†‡], Raja Bhattacharya[1†§], Michele L Lemons[2], Jeremy Florman[1], Christopher M Lambert[1], Denis Touroutine[1], Kellianne Alexander[1], Liliane Schoofs[3], Mark J Alkema[1], Isabel Beets[3], Michael M Francis[1]*

[1]Department of Neurobiology, University of Massachusetts Chan Medical School, Worcester, United States; [2]Department of Biological and Physical Sciences, Assumption University, Worcester, United States; [3]Department of Biology, University of Leuven (KU Leuven), Leuven, Belgium

*For correspondence:
michael.francis@umassmed.edu

†These authors contributed equally to this work

Present address: ‡Department of Microbiology, Immunology, and Molecular Genetics, University of California, Los Angeles, Los Angeles, United States; §Amity Institute of Biotechnology, Amity University Kolkata, West Bengal, India

Competing interest: The authors declare that no competing interests exist.

**Abstract** Neuromodulators promote adaptive behaviors that are often complex and involve concerted activity changes across circuits that are often not physically connected. It is not well understood how neuromodulatory systems accomplish these tasks. Here, we show that the *Caenorhabditis elegans* NLP-12 neuropeptide system shapes responses to food availability by modulating the activity of head and body wall motor neurons through alternate G-protein coupled receptor (GPCR) targets, CKR-1 and CKR-2. We show *ckr-2* deletion reduces body bend depth during movement under basal conditions. We demonstrate CKR-1 is a functional NLP-12 receptor and define its expression in the nervous system. In contrast to basal locomotion, biased CKR-1 GPCR stimulation of head motor neurons promotes turning during local searching. Deletion of *ckr-1* reduces head neuron activity and diminishes turning while specific *ckr-1* overexpression or head neuron activation promote turning. Thus, our studies suggest locomotor responses to changing food availability are regulated through conditional NLP-12 stimulation of head or body wall motor circuits.

## Editor's evaluation

In this work, Ramachandran and colleagues investigate how the *C. elegans* cholecystokinin-like neuropeptide (NLP-12) signaling pathway modulates animal posture during locomotion. They show that control over head- versus body-bending diverges at the level of two different NLP-12 receptors and that this fine-tuning enables the animal to reach different behavioral goals i.e., local exploration versus long-distance traveling during food search.

## Introduction

Neuromodulators serve critical roles in altering the functions of neurons to elicit alternate behavior. Disruptions in neuromodulatory transmitter systems are associated with a variety of behavioral and neuropsychiatric conditions, including eating disorders, anxiety, stress and mood disorders, depression, and schizophrenia (*Bailer and Kaye, 2003*; *Kormos and Gaszner, 2013*; *Pomrenze et al., 2019*). To achieve their effects, neuromodulatory systems may act broadly through projections across many brain regions or have circuit-specific actions, based on the GPCRs involved and their cellular expression. A single neuromodulator may therefore perform vastly different signaling functions across the circuits where it is released. For example, Neuropeptide Y (NPY) coordinates a variety of energy

and feeding-related behaviors in mammals through circuit-specific mechanisms. NPY signaling may increase or decrease food intake depending upon the circuit and GPCR targets involved (*West and Roseberry, 2017*; *Zhang et al., 2019*). Due to the varied actions of neuromodulators across cell types and neural circuits, it has remained challenging to define how specific neuromodulatory systems act in vivo to elicit alternate behaviors. Addressing this question in the mammalian brain is further complicated by the often widespread and complex projection patterns of neuromodulatory transmitter systems, and our still growing knowledge of brain connectivity.

The compact neural organization and robust genetics of invertebrate systems such as *Caenorhabditis elegans* are attractive features for studies of neuromodulatory function. Prior work has shown that *C. elegans* NLP-12 neuropeptides are key modulatory signals in the control of behavioral adaptations to changing environmental conditions, such as food availability or oxygen abundance (*Bhattacharya et al., 2014*; *Hums et al., 2016*; *Oranth et al., 2018*). The NLP-12 system is the closest relative of the mammalian Cholecystokinin (CCK) neuropeptide system and is highly conserved across flies, worms, and mammals (*Janssen et al., 2009*; *Janssen et al., 2008*; *Peeters et al., 2012*). CCK is abundantly expressed in the mammalian brain; however, a clear understanding of the regulatory actions of CCK on the circuits where it is expressed is only now beginning to emerge (*Ballaz, 2017*; *Lee and Soltesz, 2011*; *Nishimura et al., 2015*; *Saito et al., 1980*). Like mammals, the *C. elegans* genome encodes two putative CCK-responsive G protein-coupled receptors (GPCRs) (CKR-1 and CKR-2), though, prior to the present study, direct activation by NLP-12 peptides had only been demonstrated for the CKR-2 GPCR (*Frooninckx et al., 2012*; *Janssen et al., 2009*; *Janssen et al., 2008*; *Peeters et al., 2012*). The experimental tractability of *C. elegans*, combined with the highly conserved nature of the NLP-12/CCK system, offers a complementary approach for uncovering circuit-level actions underlying neuropeptide modulation, in particular, NLP-12/CCK neuropeptide signaling.

Sudden decreases in food availability or environmental oxygen levels each evoke a characteristic behavioral response in *C. elegans* where animals limit their movement to a restricted area by increasing the frequency of trajectory changes (reorientations), a behavior known as local or area-restricted searching (ARS) (*Bhattacharya et al., 2014*; *Gray et al., 2005*; *Hills et al., 2004*; *Hums et al., 2016*; *Oranth et al., 2018*). ARS is a highly conserved adaptive behavior and is evident across diverse animal species (*Bailey et al., 2019*; *Bell, 1990*; *Marques et al., 2020*; *Paiva et al., 2010*; *Sommerfeld et al., 2013*; *Weimerskirch et al., 2007*). ARS responses during food searching in particular are rapid and transient. Trajectory changes increase within a few minutes after food removal, and decrease with prolonged removal from food (>15–20 min) as animals transition to global searching (dispersal) (*Bhattacharya et al., 2014*; *Calhoun et al., 2014*; *Gray et al., 2005*; *Hills et al., 2004*; *Hums et al., 2016*; *Oranth et al., 2018*; *Wakabayashi et al., 2004*). The clearly discernible behavioral states during food searching present a highly tractable model for understanding the contributions of specific neuromodulatory systems. NLP-12 neuropeptide signaling promotes increases in body bending amplitude and turning during movement (*Bhattacharya et al., 2014*; *Hums et al., 2016*), motor adaptations that are particularly relevant for ARS. Notably, *nlp-12* is strongly expressed in only a single neuron, the interneuron DVA that has synaptic targets in the motor circuit and elsewhere (*Bhattacharya et al., 2014*; *White et al., 1997*). Despite the restricted expression of *nlp-12*, there remains considerable uncertainty about the cellular targets of NLP-12 peptides and the circuit-level mechanisms by which NLP-12 modulation promotes its behavioral effects.

Here, we explore the GPCR and cellular targets involved in NLP-12 neuromodulation of local food searching. Our findings reveal a primary requirement for NLP-12 signaling onto SMD head motor neurons, mediated through the CKR-1 GPCR, for trajectory changes during local searching. In contrast, NLP-12 signaling through both CKR-1 and CKR-2 GPCRs contribute to NLP-12 regulation of basal locomotion, likely through signaling onto head and body wall motor neurons. Our results suggest a model where NLP-12 signaling acts through CKR-1 and CKR-2 to coordinate activity changes across head and body wall motor circuits during transitions between basal and adaptive motor states.

# Results

## NLP-12/CCK induced locomotor responses require functional CKR-1 signaling

To decipher mechanisms underlying NLP-12 regulation of local food searching, we sought to identify genes required for NLP-12-mediated locomotor changes, in particular, the G protein-coupled receptors (GPCRs) responsible for NLP-12 signaling. The *C. elegans* genome encodes closely related CKR-1 and CKR-2 (Cholecystokinin-like Receptors 1 and 2) GPCRs with sequence homology to the mammalian Cholecystokinin receptors CCK-1 and CCK-2 (*Figure 1—figure supplement 1A-B*; *Janssen et al., 2009*; *Janssen et al., 2008*; *Peeters et al., 2012*). Prior work demonstrated that NLP-12 activates CKR-2 in vitro (*Janssen et al., 2008*). Further, genetic studies provided evidence that NLP-12 signaling mediates functional plasticity at cholinergic neuromuscular synapses through CKR-2 modulation of acetylcholine release from motor neurons (*Bhattacharya et al., 2014*; *Hu et al., 2015*; *Hu et al., 2011*). Surprisingly, however, deletion of *ckr-2* does not strongly affect local search behavior (*Bhattacharya et al., 2014*). As functional roles for the CKR-1 GPCR have not been previously described, we sought to determine whether CKR-1 may be acting either alone or in combination with CKR-2 to direct NLP-12 regulation of local searching. We first isolated a full-length *ckr-1* cDNA identical to the predicted *ckr-1* sequence. As expected, we found the *ckr-1* locus encodes a predicted protein containing seven transmembrane domains and sharing strong similarity to the CCK-like GPCR family (*Figure 1—figure supplement 1*).

To define potential roles for CKR-1 and CKR-2 in local searching, we took advantage of a strain we had previously generated that stably expresses high levels of the NLP-12 precursor [*nlp-12*(OE)] (*Bhattacharya et al., 2014*). Overexpression of *nlp-12* in this manner elicits exaggerated loopy movement, increased trajectory changes, and enhanced body bend amplitude (*Figure 1A*, Figure 6C, *Video 1*). The average amplitude of bending is increased approximately threefold in comparison to wild type (*Figure 1B*), and body bends are more broadly distributed over steeper angles (*Figure 1C–D*). These overexpression effects are constitutive, offering experimental advantages for pursuing genetic strategies to identify signaling mechanisms. We investigated the requirement for CKR-1 and CKR-2 in the locomotor changes elicited by *nlp-12* overexpression using available strains carrying independent deletions in each of these genes. The *ckr-2* deletion (*tm3082*) has been characterized previously and likely represents a null allele (*Hu et al., 2011*; *Janssen et al., 2008*; *Peeters et al., 2012*). The *ckr-1* deletion (*ok2502*) removes 1289 base pairs, including exons 3–7 that encode predicted transmembrane domains 2–5 (*Figure 1—figure supplement 1B-C*) and therefore also likely represents a null allele. *ckr-1* and *ckr-2* single gene deletions each partially reversed the effects of *nlp-12* overexpression (*Figure 1A,B,D*, 6C), indicating that both CKR-1 and CKR-2 GPCRs are active under conditions when NLP-12 peptides are present at high levels. Notably, *ckr-1* deletion showed slightly greater suppression of *nlp-12(OE)* phenotypes compared with *ckr-2* deletion (*Figure 1B,D*, 6C). Combined deletion of *ckr-1* and *ckr-2* largely reversed the locomotor changes produced by NLP-12 overexpression (*Figure 1A,B,D*, 6C), indicating that the GPCRs act in a partially redundant manner. Our genetic analysis of *nlp-12* overexpression confirms a role for the CKR-2 GPCR in NLP-12-elicited motor adaptations, and importantly, provides first evidence implicating the previously uncharacterized CKR-1 GPCR in NLP-12 modulation of motor activity.

## NLP-12 activates CKR-1 with high potency

To obtain direct evidence for NLP-12 activation of CKR-1, we used an in vitro bioluminescence-based approach. CKR-1 was expressed in Chinese hamster ovarian (CHO) cells stably expressing the promiscuous G-protein alpha subunit $G_{\alpha16}$ and a bioluminescent calcium indicator, aequorin (*Caers et al., 2014*). The NLP-12 precursor gives rise to two distinct mature peptides, NLP-12–1 and NLP-12–2. Application of either NLP-12–1 or NLP-12–2 synthetic peptides produced robust calcium responses in cells expressing CKR-1. These responses were concentration-dependent with $EC_{50}$ values of 3.5 and 1.9 nM for NLP-12–1 and NLP-12–2 peptides, respectively (*Figure 1E*). These $EC_{50}$ values are comparable to those measured for NLP-12 activation of CKR-2 (8.0 nM and 10.2 nM) (*Figure 1F*; *Janssen et al., 2008*), suggesting NLP-12 peptides act with similar potency across CKR-1 and CKR-2 GPCRs. Importantly, no other peptides from a library of over 350 synthetic *C. elegans* peptides elicited CKR-1 activation, nor did the NLP-12 peptides evoke calcium responses in cells transfected with

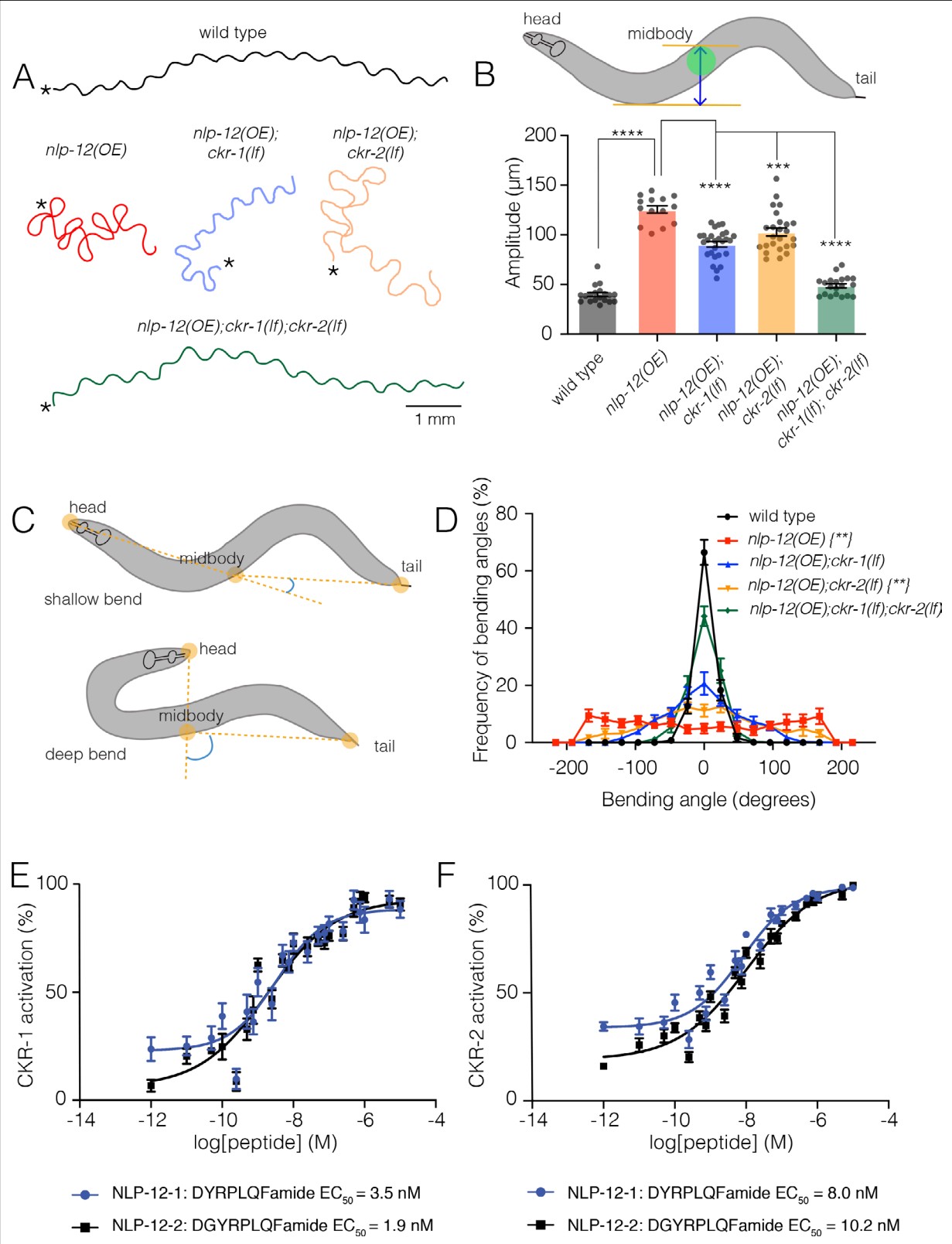

**Figure 1.** NLP-12/CCK induced locomotor responses require functional *ckr-1* signaling. (**A**) Representative movement trajectories of wild-type (black), *nlp-12(OE)* (red), *nlp-12(OE);ckr-1(lf)* (blue), *nlp-12(OE);ckr-2(lf)* (orange), and *nlp-12(OE);ckr-1(lf);ckr-2(lf)* (green) animals during forward runs (30 s) on NGM agar plates seeded with OP50 bacteria. *nlp-12(OE)* refers to the transgenic strain (*ufIs104*) stably expressing high levels of wild-type *nlp-12* genomic sequence. Note the convoluted *nlp-12(OE)* movement tracks are restored to wild type by combined *ckr-1* and *ckr-2* deletion. Scale bar, 1 mm.

*Figure 1 continued on next page*

**Figure 1 continued**

Asterisks (*) indicate position of worm at start of recording. (**B**) Average body bend amplitude (indicated in schematic by blue arrow between orange lines, midbody centroid [green] of worm) for the genotypes as indicated. Bars represent mean ± SEM. In this and subsequent figures. ****p<0.0001, ***p<0.001, ANOVA with Holms-Sidak post hoc test. wild-type n=19, *nlp-12(OE)*: n=14, *nlp-12(OE);ckr-1(lf)*: n=27, *nlp-12(OE);ckr-2(lf)*: n=25, *nlp-12(OE);ckr-1(lf);ckr-2(lf)*: n=20. (**C**) Schematic representation of measured body bending angle, for shallow (top) and deep (bottom) body bends. Solid orange circles indicate the vertices (head, midbody, and tail) of the body bending angle (blue) measured. (**D**) Frequency distribution of body bending angle (indicated in blue in (**C**)) for the genotypes indicated. Kolmogorov-Smirnov test: wild-type versus *nlp-12(OE)*\*\*, wild-type versus *nlp-12(OE);ckr-2(lf)*\*\*, *nlp-12(OE) versus nlp-12(OE);ckr-1(lf);ckr-2(lf)*\*\*, \*\*p<0.01. wild-type: n=12, *nlp-12(OE)*: n=10, *nlp-12(OE);ckr-1(lf)*: n=10, *nlp-12(OE);ckr-2(lf)*: n=12, *nlp-12(OE);ckr-1(lf);ckr-2(lf)*: n=12. (**E, F**) Concentration-response curves of the mean calcium responses (% activation ± SEM) in CHO cells expressing either CKR-1 (**E**) or CKR-2 (**F**) for different concentrations of synthetic peptides NLP-12–1 (solid blue circles) or NLP-12–2 (solid black squares). Solid lines indicate curve fits to the data (n=6). 95% confidence intervals (nM), CKR-1: NLP-12–1, 1.79–7.07; NLP-12–2, 0.93–3.77 and CKR-2: NLP-12–1, 5.16–12.51; NLP-12–2, 6.43–16.73. NGM, nematode growth media.

The online version of this article includes the following figure supplement(s) for figure 1:

**Source data 1.** Source data for body bending amplitude (*Figure 1B*).

**Source data 2.** Source data for frequency of bending angles (*Figure 1D*).

**Source data 3.** Source data for in vitro analysis of CKR-1 activation (*Figure 1E*).

**Source data 4.** Source data for in vitro analysis of CKR-2 activation (*Figure 1F*).

**Figure supplement 1.** CKR-1 and CKR-2 GPCRs share similarity with vertebrate CCK GPCRs.

**Figure supplement 2.** NLP-12 peptides activate CKR-1 and CKR-2 in vitro.

**Figure supplement 2—source data 1.** Source data for in vitro controls (ratio of total calcium response).

empty vector (*Figure 1—figure supplement 2*), indicating that CKR-1, like CKR-2, is a highly specific receptor for NLP-12.

## CKR-1 is a key signaling component for local search behavior

To more deeply investigate roles for CKR-1 and CKR-2 in NLP-12 regulation of movement, we quantified body and head bending during basal locomotion (in the presence of food) using single worm tracking analysis. *nlp-12* deletion significantly reduced both body bending and head bending angles in comparison to wild type (*Figure 2A–B*). Similarly, single deletions in *ckr-1* and *ckr-2* each produced significant reductions in body bending, and combined deletion produced effects similar to *nlp-12* deletion (*Figure 2A*). In contrast, head bending was strikingly affected by *ckr-1* deletion, while *ckr-2* deletion did not produce a significant reduction (*Figure 2B*). The preferential involvement of CKR-1 in head bending suggested the interesting possibility that CKR-1 and CKR-2 GPCRs differentially regulate specific features of locomotion.

To explore this possibility further, we investigated the involvement of CKR-1 and CKR-2 GPCRs in local search responses following removal from food. Specifically, we monitored worm movement during a 35-min period immediately after removal from food and quantified turning behavior during the first (0–5, local searching, *Video 2*) and last (30–35, dispersal, *Video 3*) five minutes (*Figure 3A*). Post hoc video analysis proved most reliable for measuring turning behavior during local searching. We quantified changes in trajectory (reorientations), that resulted in a change of >50° in the direction of movement, executed either through forward turns or reversal-coupled omega turns (*Figure 3B*, *Figure 3—figure supplement 1*). For wild type, we noted an increase in reorientations immediately following removal from food compared to animals maintained on food (*Figure 3—figure supplement 2A*). Consistent with our previous findings (*Bhattacharya et al., 2014*), we found that deletion of *nlp-12* significantly decreased reorientations immediately following removal from food (*Figure 3C–D*).

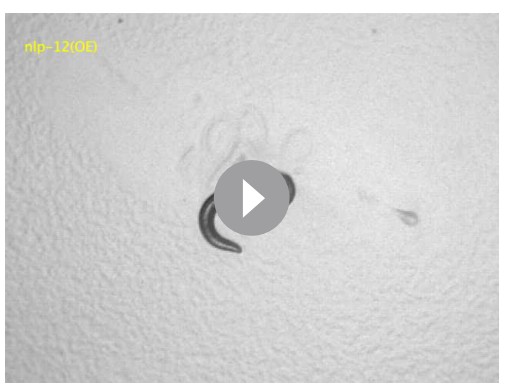

**Video 1.** Representative 20-s video showing locomotion on food of animal overexpressing *nlp-12*. Video has been sped up 4×.

https://elifesciences.org/articles/71747/figures#video1

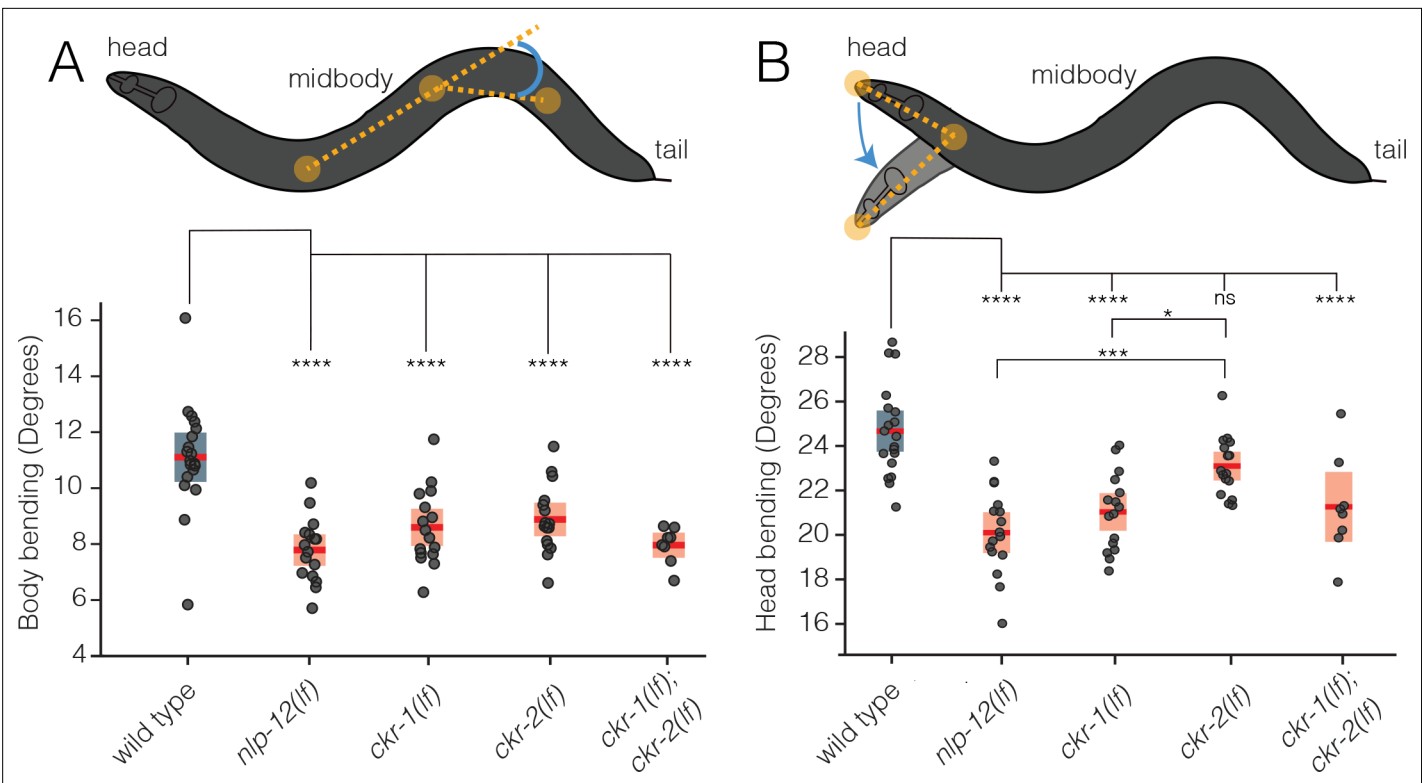

**Figure 2.** CKR-1 and CKR-2 differentially regulate head and body bending during basal locomotion. Schematics showing body bending (**A**) and head bending (**B**) angles (solid orange circles indicate the vertices and measured angle in blue) quantified during single worm track analyses of movement (5 min) in the presence of food. Each data point in the scatterplots represents the average body or head bend angle for a single animal from analysis of 5 min of locomotion. Horizontal red bar indicates mean, shading indicates SEM for wild-type (blue) and mutants (orange). ****$p<0.0001$, ***$p<0.001$, *$p<0.05$, ns, not significant. ANOVA with Holms-Sidak post hoc test. wild-type: n=19, *nlp-12(ok335)*: n=16, *ckr-1(ok2502)*: n=16, *ckr-2(tm3082)*: n=16, *ckr-1(ok2502);ckr-2(tm3082)*: n=8.

The online version of this article includes the following figure supplement(s) for figure 2:

**Source data 1.** Source data for body bending measurements during single worm tracking of basal locomotion (*Figure 2A*).

**Source data 2.** Source data for head bending measurements during single worm tracking of basal locomotion (*Figure 2B*).

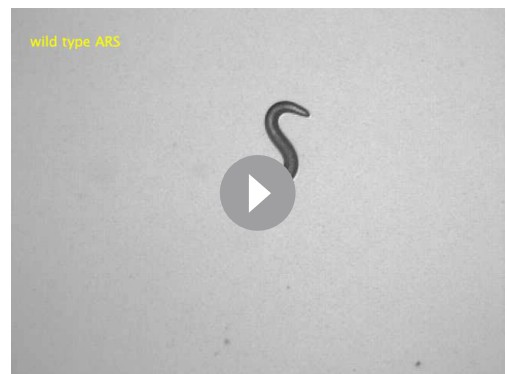

**Video 2.** Representative 20-s video showing locomotion of wild-type animal during area restricted search (0–5 min off food). Video has been sped up 4×.
https://elifesciences.org/articles/71747/figures#video2

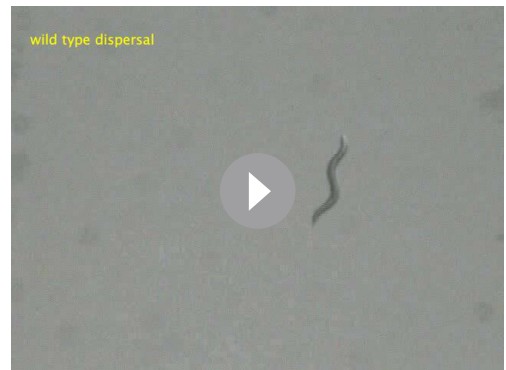

**Video 3.** Representative 20-s video showing locomotion of wild-type animal during dispersal (30–35 mi off food). Video has been sped up 4×.
https://elifesciences.org/articles/71747/figures#video3

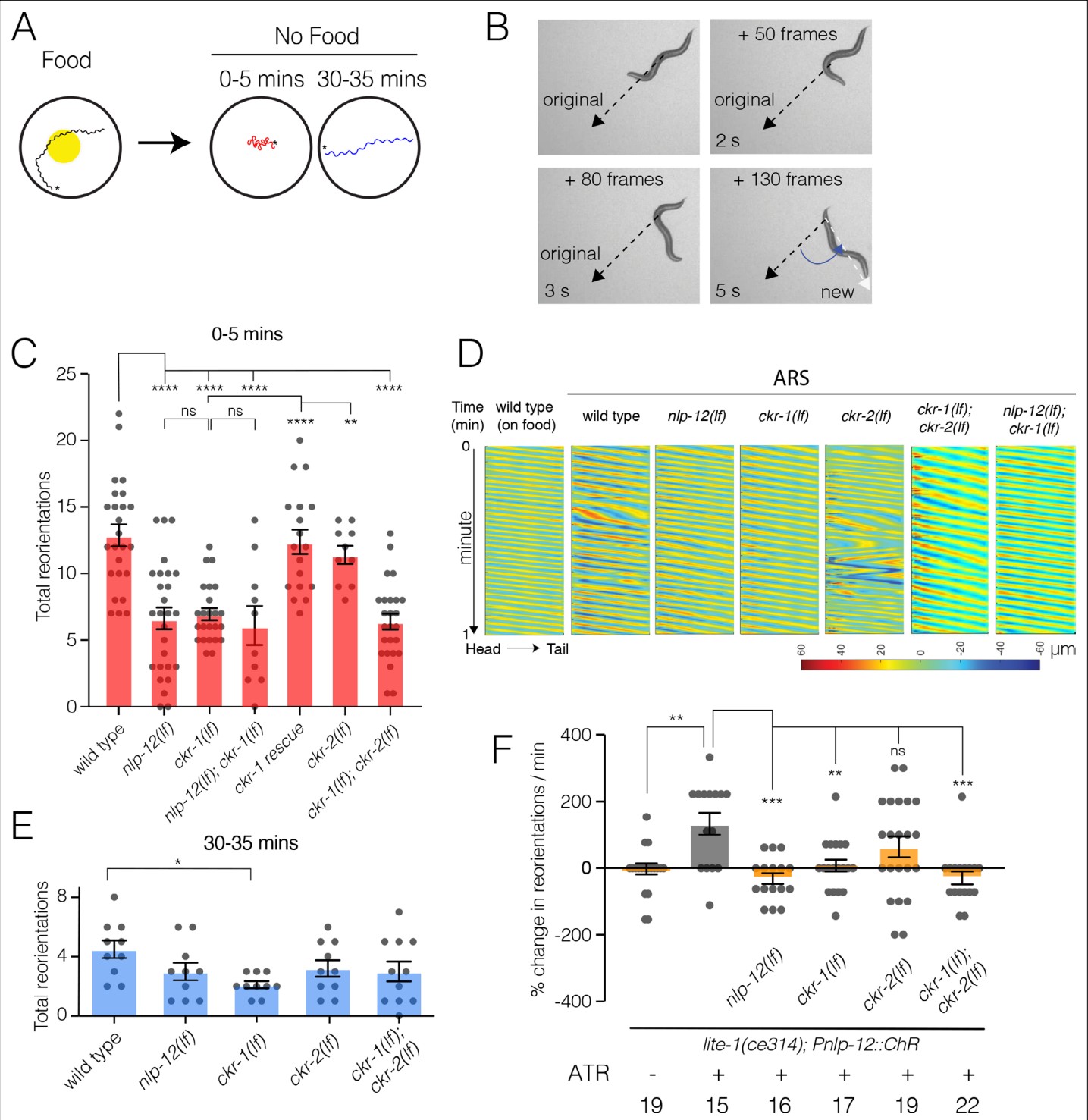

**Figure 3.** NLP-12/CCK food search responses are mediated through the GPCR CKR-1. (**A**) Schematic of the food search assay indicating the time intervals when reorientations were scored. Wild-type animals increase reorientations during the first 5 min (0–5 min) after removal from food (local search) and reduce reorientations during dispersal (30–35 min). Asterisks (*) indicate the position of worm at the start of recording. (**B**) Frame grabs showing worm position and posture prior to, during and after reorientation. Angle (blue) between the black (original trajectory) and white (new trajectory) dashed lines indicates the change in trajectory. Frame numbers and time points indicated are relative to the first image in each sequence, which represents the start point (frame 0, time 0 s) when the reorientation event began, and the last frame was when the reorientation was completed. Trajectory changes were scored as reorientations if changes in trajectory were greater than 50°. (**C**) Quantification of reorientations during 0–5 min following removal from food for the genotypes indicated. Rescue refers to transgenic expression of wild-type *ckr-1* in *ckr-1* mutants. Bars represent

*Figure 3 continued on next page*

*Figure 3 continued*

mean ± SEM. \*\*\*\*p<0.0001, \*\*p<0.01, ns, not significant, ANOVA with Holms-Sidak post hoc test. wild-type: n=25, *nlp-12(ok335)*: n=27, *ckr-1(ok2502)*: n=24, *nlp-12(ok335);ckr-1(ok2502)*: n=10, *ckr-1* rescue: n=18, *ckr-2(tm3082)*: n=10, *ckr-1(ok2502);ckr-2(tm3082)*: n=25. (**D**) Representative body curvature kymographs for worm locomotion during basal locomotion and area restricted searching (ARS). Head to tail orientation along the horizontal axis in each kymograph is left to right as indicated for wild type. Time is indicated along the vertical axis from 0 min to 1 min. (**E**) Total number of reorientations during an interval of 30–35 min following removal from food for the genotypes as shown. Each bar represents mean ± SEM. \*p<0.05, ANOVA with Holms-Sidak post hoc test. wild-type: n=10, *nlp-12(ok335)*: n=10, *ckr-1(ok2502)*: n=10, *ckr-2(tm3082)*: n=10, *ckr-1(ok2502);ckr-2(tm3082)*: n=11. (**F**) Trajectory changes (reorientations) scored in response to photostimulation of DVA. Percent change in the number of high angle turns elicited during 1 min of blue light exposure compared to prestimulus (no blue light). Bars represent mean ± SEM. \*\*\*p<0.001, \*\*p<0.01, ns, not significant, compared to +ATR control, ANOVA with Holms-Sidak post hoc test. ATR, all-trans retinal.

The online version of this article includes the following figure supplement(s) for figure 3:

**Source data 1.** Source data for reorientations quantified during area restricted search (0–5 min off food, *Figure 3C*).

**Source data 2.** Source data for reorientations quantified during dispersal (30–35 min off food, *Figure 3E*).

**Source data 3.** Source data for % change in reorientations from mean quantified for DVA photostimulation (*Figure 3F*).

**Figure supplement 1.** Sequential snapshots of frames from a representative reorientation, for forward reorientations (**A**) and reversal-coupled omega turn mediated reorientations (**B**).

**Figure supplement 2.** NLP-12 signaling through CKR-1 promotes forward reorientations.

**Figure supplement 2—source data 1.** Source data for reorientations quantified on food and during area restricted search (0–5 min off food, *Figure 3—figure supplement 2A*).

**Figure supplement 2—source data 2.** Source data for reorientations quantified during area restricted search (0–5 min off food, *Figure 3—figure supplement 2B*).

**Figure supplement 3.** NLP-12 released from DVA acts selectively through CKR-1 to promote reorientations.

**Figure supplement 3—source data 1.** Source data for reorientations quantified during area restricted search (0–5 min off food, *Figure 3—figure supplement 3A*).

**Figure supplement 3—source data 2.** Source data for reorientations quantified during area restricted search (0–5 min off food, *Figure 3—figure supplement 3B*).

**Figure supplement 3—source data 3.** Source data for reorientations quantified during area restricted search (0–5 min off food, *Figure 3—figure supplement 3C*).

In particular, we noted a significant reduction in the forward reorientations of *nlp-12* mutants, but no appreciable effect on reversal-coupled omega turns (*Figure 3—figure supplement 2B*). Deletion of *ckr-2* produced no appreciable effect on reorientations (*Figure 3C–D*; *Bhattacharya et al., 2014*); however, single deletion of *ckr-1* decreased reorientations to a similar level as observed for *nlp-12* deletion (*Figure 3C–D*). Similar to *nlp-12(lf)*, we found that *ckr-1(lf)* significantly impacted forward reorientations, but did not affect reversal-coupled omega turns (*Figure 3—figure supplement 2B*). Combined deletion of *ckr-1* and *ckr-2* provided no additional decrease beyond that observed for single *ckr-1* deletion (*Figure 3C–D*). In addition, combined deletion of *nlp-12* and *ckr-1* did not further decrease reorientations compared with either of the single mutants (*Figure 3C–D*). Expression of wild-type *ckr-1*, but not *ckr-2*, rescued reorientations in *ckr-1(lf);ckr-2(lf)* double mutants (*Figure 3—figure supplement 3A*). Expression of wild-type *ckr-1* also restored normal reorientation behavior in *ckr-1(lf)* animals when expressed under the control of native *ckr-1* promoter elements (3.5 kb) (*Figure 3C*), but not when expressed under the *ckr-2* promoter (*Figure 3—figure supplement 3B*). These findings show that *nlp-12* and *ckr-1* act in the same genetic pathway and point to a selective requirement for NLP-12 signaling through CKR-1 in regulating trajectory changes during local searching. Deletion of *nlp-12* did not produce significant changes in dispersal behavior, but we noted a modest decrease in reorientations during dispersal in *ckr-1* mutants (*Figure 3E*). This may indicate additional roles for CKR-1 during dispersal. Taken together, our genetic and behavioral studies implicate CKR-1 and CKR-2 GPCRs as targets of NLP-12 signaling under conditions of overexpression and during basal locomotion. In contrast, we find that NLP-12 modulation of local searching is primarily achieved through CKR-1 activation.

## Acute stimulation of DVA promotes reorientation behavior and requires NLP-12 and CKR-1

We next addressed the question of how neuronal release of NLP-12 promotes area restricted searching. We measured trajectory changes elicited by acute depolarization of the DVA neuron. We used the *nlp-12* promoter to drive cell-specific expression of Channelrhodopsin-2 (ChR2) (*Nagel et al., 2003*) in DVA and tracked worm movement during a 1-min period of blue light (470 nm) photostimulation. We found that animals reorient more frequently with depolarization of DVA compared to pre-stimulus control (*Figure 3F*). Importantly, light exposure did not increase reorientations in the absence of retinal (–ATR) (*Figure 3F*). Depolarization of the DVA neuron in *nlp-12* mutants failed to produce a similar enhancement (*Figure 3F*), offering support for the idea that reorientations primarily arise due to the release of NLP-12 peptides. Single *ckr-1* deletion or combined *ckr-1* and *ckr-2* deletion also abrogated DVA-elicited increases in reorientation behavior, while single *ckr-2* deletion produced more variable responses that were not clearly distinguishable from control (*Figure 3F*). Our photostimulation experiments provide direct evidence that NLP-12 release from the DVA neuron promotes reorientation behavior, and, in addition, provide evidence for central involvement of NLP-12 signaling through the CKR-1 GPCR in directing reorientations. While NLP-12 expression has also been recently reported in PVD neurons (*Tao et al., 2019*), expression of *nlp-12* under a PVD specific promoter (*ser-2prom3*) did not restore reorientations in *nlp-12(lf)* animals (*Figure 3—figure supplement 3C*), pointing toward DVA as the primary source of NLP-12 in promoting reorientations.

## Elevated CKR-1 signaling enhances turning and body bending in an *Nlp-12* dependent manner

To further define the role of CKR-1, we next asked whether increased CKR-1 signaling would be sufficient to induce local search-like behavior. To address this question, we pursued an overexpression strategy similar to our above approach for *nlp-12*. We generated transgenic lines where the *ckr-1* genomic sequence including native *ckr-1* promoter elements was injected into wild-type animals at high concentration.

We found that *ckr-1* overexpression produced striking increases in turning and large head to tail body bends (*Figure 4A*, 6C, *Video 4*), qualitatively similar to the effects of *nlp-12* overexpression (*Figure 1A*, *Video 1*). *ckr-1*(OE) animals made steep bends during runs of forward movement, with angles approaching 200°, whereas bending angles in wild type rarely exceeded 75° (*Figure 4B*). Notably, these high angle bends often produced spontaneous reorientations during forward movement and sometimes elicited sustained coiling. The amplitude of body bends during movement also increased by approximately threefold in *ckr-1*(OE) animals compared to wild type (*Figure 4C*). These increases in bending angles and body bend depth were returned to wild-type levels by *nlp-12* deletion (*Figure 4A–C*), offering support that NLP-12 peptides are the major CKR-1 ligands required to elicit these characteristic changes in movement. Taken together, our genetic studies define NLP-12/CKR-1 as a novel ligand-GPCR pathway that controls trajectory changes and body bending to produce adaptive behavior.

## *ckr-1* is expressed in many neurons that do not receive direct synaptic inputs from DVA

To identify cells where CKR-1 may act to promote local searching, we generated strains expressing a *ckr-1* reporter transgene that included the complete *ckr-1* genomic locus and ~3.5 kb of upstream regulatory sequence SL2 trans-spliced to sequence encoding GFP (green fluorescent protein) or mCherry. We found that *ckr-1* is broadly expressed in the nervous system, showing expression in a subset of ventral nerve cord motor neurons, amphid and phasmid sensory neurons, premotor interneurons, and motor neurons in the nerve ring (*Figure 5A–B*). We identified many of these neurons, largely from analysis of *ckr-1* co-expression with previously characterized reporters (*Supplementary file 2*). In the ventral nerve cord, we found that *ckr-1* is expressed in cholinergic, but not GABAergic, ventral cord motor neurons (*Figure 5—figure supplement 1A-B*, *Supplementary file 2*). Amongst head neurons, the *ckr-1* reporter is expressed in GABAergic RMEV, RMED, AVL and RIS neurons, cholinergic SMDV, SMDD, and RIV head motor neurons, the interneuron RIG, the serotonergic NSM neuron, and in the interneurons AIA and AIB (*Figure 5B*, *Supplementary file 2*). Additional studies using DiI uptake indicated that *ckr-1* is also expressed in the amphid sensory neurons ASK and ASI and

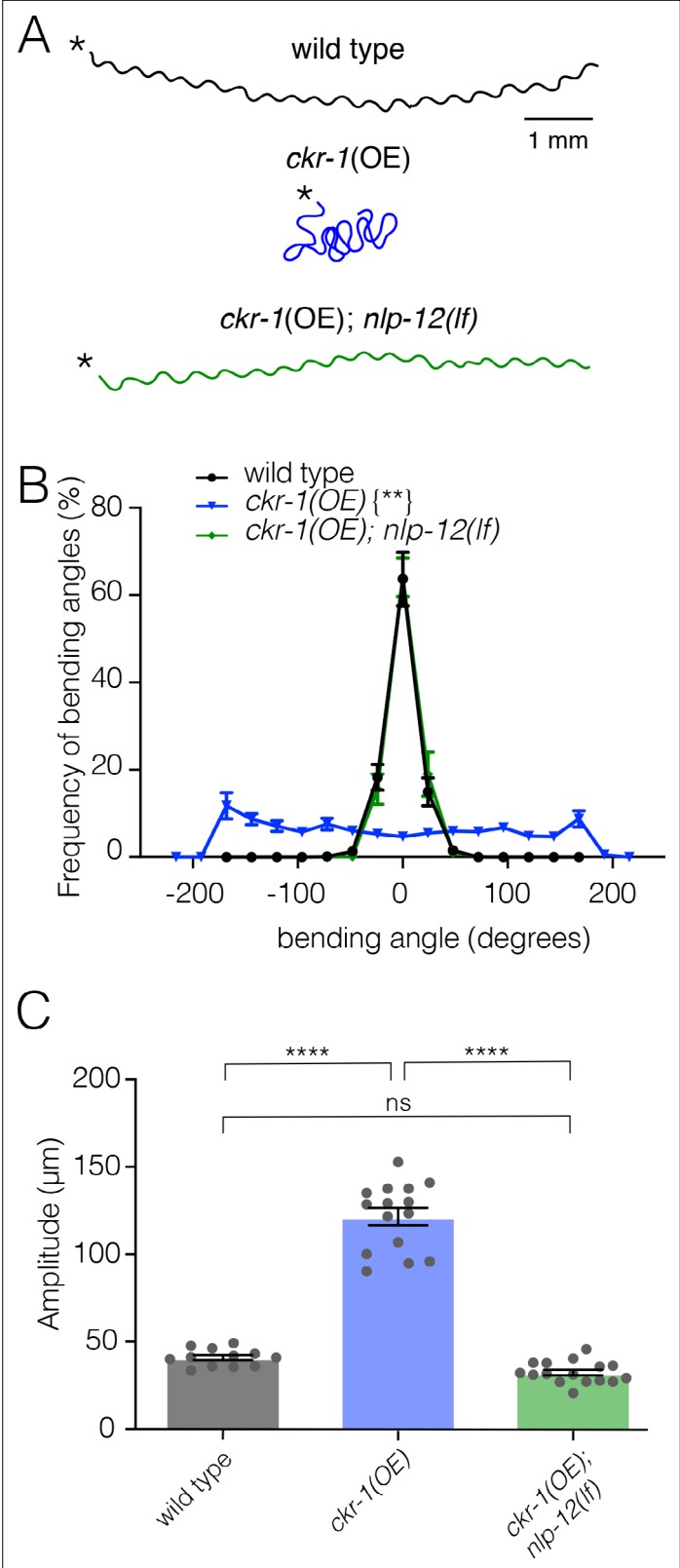

**Figure 4.** Elevated CKR-1 signaling enhances bending angle and amplitude in an *nlp-12* dependent manner. (**A**) Representative movement trajectories of wild-type (black), *ckr-1(OE)* (blue) and *ckr-1(OE); nlp-12(lf)* (green) animals for 30 s on NGM agar plates seeded with OP50 bacteria. *ckr-1(OE)* refers to high copy expression of the wild-type *ckr-1* genomic locus (*ufEx802*). Note the increased frequency of high angle turns and convoluted track for *ckr-*

*Figure 4 continued on next page*

*Figure 4 continued*

*1(OE)*. These movement phenotypes are reversed by *nlp-12* deletion. Scale bar, 1 mm. (**B**) Frequency distribution of body bending angles (mean ± SEM) during forward runs (30 s) on plates thinly seeded with OP50 bacteria. Kolmogorov-Smirnov test: wild-type versus *ckr-1(OE)*\*\*, *ckr-1(OE)* versus *ckr-1(OE); nlp-12(ok335)*\*\*, wild-type versus *ckr-1(OE); nlp-12(ok335)* ns. \*\*p<0.01, ns, not significant. wild-type: n=8, *ckr-1(OE)*: n=10, and *ckr-1(OE);nlp-12(lf)*: n=10. (**C**) Comparison of the average body bend amplitude for the indicated genotypes. Bars represent mean ± SEM. \*\*\*\*p<0.0001, ns, not significant, ANOVA with Holms-Sidak post hoc test. wild-type: n=12, *ckr-1(OE)*: n=15, *ckr-1(OE);nlp-12(ok335)*: n=16. NGM, nematode growth media.

The online version of this article includes the following figure supplement(s) for figure 4:

**Source data 1.** Source data for frequency of bending angles (*Figure 4B*).

**Source data 2.** Source data for body bending amplitude (*Figure 4C*).

the phasmid sensory neurons PHA and PHB (*Supplementary file 2*). With the exception of the ventral cord cholinergic neurons, the *ckr-1* reporter almost exclusively labeled neurons that do not receive direct synaptic input from DVA, suggesting that NLP-12 acts at least partially through extrasynaptic mechanisms. Notably, *ckr-1* and *ckr-2* expression showed little overlap (*Figure 5—figure supplement 2*).

## CKR-1 functions in the SMD head motor neurons to modulate body bending

We next pursued cell-specific *ckr-1* overexpression to gain insight into which *ckr-1*-expressing neurons defined above may be primary targets for modulation during local searching (*Supplementary files 3-4*). We focused our analysis on body bending amplitude because this was the most easily quantifiable aspect of movement to be modified by *ckr-1* overexpression. Transgenic strains where pan-neuronally expressed *ckr-1* (*rgef-1* promoter) was injected at high concentration displayed increased body bending amplitude, similar to overexpression using the native promoter (*Figure 5C*). In contrast, ectopic *ckr-1* expression in muscles produced no appreciable change, consistent with a primary site of CKR-1 action in neurons (*Figure 5C*). Surprisingly, *ckr-1* overexpression in cholinergic (*unc-17β* promoter) or GABAergic (*unc-47* promoter) ventral nerve cord motor neurons did not elicit an appreciable change in body bend depth (*Figure 5C*). We therefore next targeted the head neurons identified by our *ckr-1* reporter, using several different promoters for *ckr-1* overexpression in subsets of head neurons (*Figure 5C*, *Supplementary files 3-4*). *ckr-1* overexpression using either the *odr-2(16)* or *lgc-55* promoters produced a striking (2.5-fold) increase in body bend depth, comparable with *ckr-1* overexpressed under its endogenous promoter. In contrast, *ckr-1* overexpression in GABAergic neurons, including RMED and RMEV (*unc-47* promoter), did not produce an appreciable effect. Likewise, *ckr-1* overexpression in RIV, RIG, NSM, AIA, AIB, or amphid neurons failed to significantly enhance body bend depth. The *lgc-55* promoter drives expression in AVB, RMD, SMD, and IL1 neurons, as well as neck muscles and a few other head neurons (*Pirri et al., 2009*), while the *odr-2(16)* promoter primarily labels the RME and SMD head neurons (*Chou et al., 2001*; *Supplementary files 2-3*). The overlapping expression of the *odr-2(16)* and *lgc-55* promoters in SMD neurons suggested that these neurons may be centrally involved. SMD co-labeling by *ckr-1::SL2::mCherry* and *Plad-2::GFP* (*Wang et al., 2008*) provided additional evidence for *ckr-1* expression in these neurons (*Figure 5—figure supplement 1C*). In contrast to *ckr-1*, *ckr-2* was either absent or more variably expressed in a subset of the SMD neurons, the SMDDs (*Figure 5—figure supplement 1D*). Intriguingly, we noted that NLP-12::Venus clusters in the nerve ring region of the DVA process (*Figure 5D*) are concentrated in the vicinity of SMD processes (*Figure 5E*).

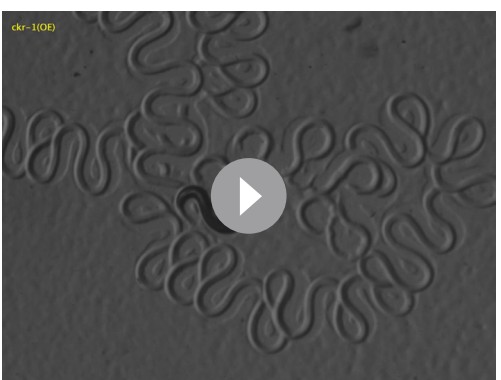

**Video 4.** Representative 20-s video showing locomotion on food of animal overexpressing *ckr-1*. Video has been sped up 4×.

https://elifesciences.org/articles/71747/figures#video4

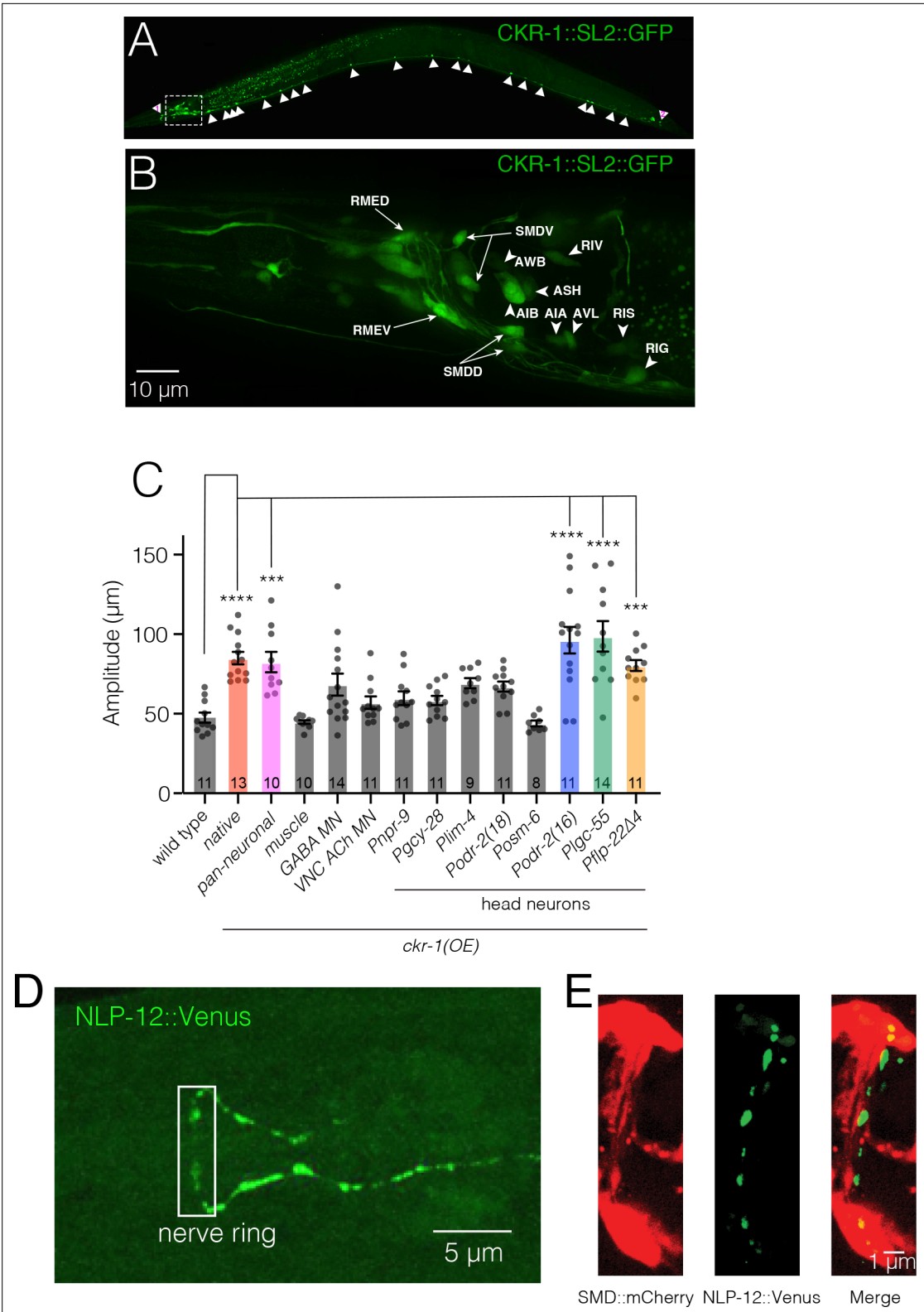

**Figure 5.** *ckr-1* functions in the SMD head motor neurons to modulate body bending. (**A**) Confocal maximum intensity projection of adult expressing the *Pckr-1::ckr-1::SL2::GFP* reporter. Note that the expression in multiple head neurons (white box) and a subset of ventral nerve cord motor neurons (white arrowheads). (**B**) Confocal maximum intensity projection of the head region of adult expressing the *Pckr-1::ckr-1::SL2::GFP* reporter. Scale bar, 10 µm. See *Figure 5—figure supplement 1* and *Supplementary file 2* for additional expression information. (**C**) Quantification of average body bend

*Figure 5 continued on next page*

Figure 5 continued

amplitudes (mean ± SEM) for *ckr-1* overexpression in the indicated cell types. Promoters used for listed cell types: pan-neuronal *Prgef-1*, muscle *Pmyo-3*, GABA motor neurons *Punc-47*, cholinergic ventral cord motor neurons *Punc-17β*. See *Supplementary file 3* for details about cellular expression of promoters used for head neurons. ****p<0.0001, ***p<0.001, ANOVA with Holms-Sidak's post hoc test. Numbers within bars indicate n for each genotype. (**D**) Confocal maximum intensity projection of the nerve ring region of a transgenic animal expressing *Pnlp-12::NLP-12::Venus*. Note the high levels of NLP-12::Venus in the nerve ring. White box indicates approximate nerve ring region where close localization of NLP-12 clusters to SMD processes has been shown in panel (**E**). Scale bar, 5 µm. (**E**) Confocal maximum intensity projection of the nerve ring region of a transgenic animal expressing *Pnlp-12::NLP-12::Venus* (DVA) and *Pflp-22Δ4::mCherry* (SMD). Note the close localization of NLP-12::Venus dense core vesicle clusters to the SMD process. Scale bar, 1 µm.

The online version of this article includes the following figure supplement(s) for figure 5:

**Source data 1.** Source data for body bending amplitude (*Figure 5C*).

**Figure supplement 1.** Neuronal expression of CKR-1 and CKR-2.

**Figure supplement 2.** CKR-1 and CKR-2 expression are largely non-overlapping.

The four SMDs (dorsal-projecting SMDDL and SMDDR and ventral-projecting SMDVL and SMDVR) are bilateral motor neuron pairs that innervate dorsal and ventral head/neck musculature, and also form reciprocal connections with one another (*White et al., 1997*). They have been previously implicated in directional head bending and steering (*Gray et al., 2005*; *Hendricks et al., 2012*; *Kaplan et al., 2020*; *Kocabas et al., 2012*; *Shen et al., 2016*; *Yeon et al., 2018*). To better define the behavioral effects of SMD modulation, we more closely examined body bending in animals overexpressing *ckr-1* under control of the *odr-2(16)* promoter, and also using a second promoter, *flp-22Δ4*, that was recently shown to drive selective expression in the SMD neurons (*Yeon et al., 2018*). For both overexpression strains, we observed significant increases in body bending amplitude and bending angle compared to wild type (*Figures 5C and 6A–C*, *Video 5*). These increases were dependent on NLP-12 signaling (*Figure 6*, *Figure 6—figure supplement 1A-B*) and were similar to those observed for native *ckr-1* (*Figures 4 and 6C*, *Video 4*) and *nlp-12* overexpression (*Figures 1 and 6C*, *Video 1*). Thus, the actions of CKR-1 in the SMD motor neurons recapitulate many of the behavioral effects of NLP-12 overexpression.

To ask if the SMD neurons are required for the locomotor changes produced by *ckr-1* overexpression, we expressed the photoactivatable cell ablation agent PH-miniSOG in the SMD neurons (P*flp-22Δ4*) of animals overexpressing *ckr-1* (native promoter). When activated by blue light (470 nm) PH-miniSOG produces reactive oxygen species and disrupts cellular function (*Xu and Chisholm, 2016*). Following photoactivation of miniSOG in animals overexpressing *ckr-1*, we observed striking decreases in bending angles (*Figure 6D–E*) and amplitude (*Figure 6F*) during movement. We confirmed successful SMD ablation by examining morphological changes in GFP-labeled SMD neurons following photoactivation of miniSOG (*Figure 6D*). Expression of miniSOG did not have appreciable effects on the body bending of *ckr-1(OE)* animals under control conditions (without light exposure) (*Figure 6—figure supplement 1C*). In addition, stimulation of control animals without the miniSOG transgene did not appreciably alter body bending (*Figure 6E*) or SMD neuron morphology (*Figure 6—figure supplement 1D*). These results indicate that SMD motor neurons are required for the locomotor effects of *ckr-1* overexpression, and, importantly, raise the possibility that the SMD neurons are key targets for NLP-12 neuromodulation during local searching in wild type.

## NLP-12/CKR-1 excitation of the SMD neurons promotes local searching

To further investigate the site of CKR-1 function, we examined rescue of area restricted searching in *ckr-1* mutants by generating additional transgenic lines providing for SMD-specific expression of wild-type *ckr-1* (injected at fivefold lower concentration than used for overexpression above). Injection of wild-type animals with the *SMD::ckr-1* transgene at this lower concentration did not appreciably increase bending depth or angle (*Figure 7—figure supplement 1A*). However, expression in *ckr-1* mutants restored reorientations during food searching to roughly wild-type levels (*Figure 7A*), indicating that CKR-1 function in the SMD neurons is sufficient to support NLP-12 modulation of local searching.

To investigate how increased SMD activity may impact movement, we photostimulated the SMDs in animals expressing P*odr-2(16)::*Chrimson (*Klapoetke et al., 2014*). Prior to photostimulation, animals demonstrated long forward runs with relatively few changes in trajectory (*Figure 7B*).

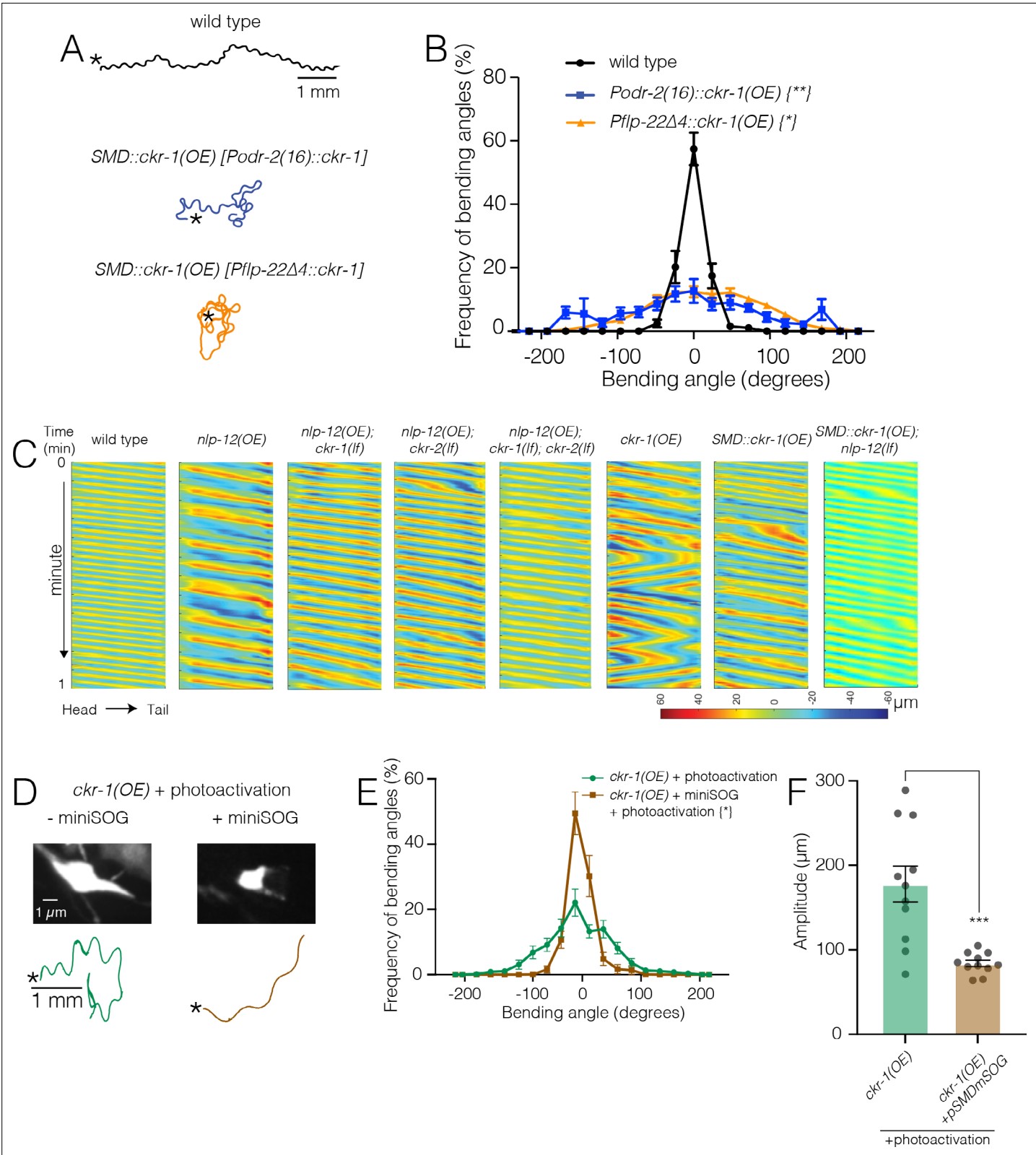

**Figure 6.** Ablation of SMD motor neurons abolishes the effects of *ckr-1* overexpression. (**A**) Representative tracks (1 min) for indicated genotypes. Asterisks indicate the position of animal at the beginning of recordings. Note that the increased reorientations and body bending depth in the tracks with cell-specific *ckr-1* overexpression. Scale bar, 1 mm. (**B**) Average body bending angle distribution (mean ± SEM) for the indicated genotypes. High level expression of *ckr-1* in SMDs using the *odr-2(16)* or *flp-22Δ4* promoters increases bending angle. Kolmogorov-Smirnov test: wild-type versus *Podr-*

*Figure 6 continued on next page*

*Figure 6 continued*

*2(16)::ckr-1(OE)*\*\*, wild-type versus *Pflp-22Δ4::ckr-1(OE)*\*, \*\*p<0.01, \*p<0.05. wild-type n=9 (black circles), *Podr-2(16)::ckr-1(OE)*: n=9 (blue squares), *Pflp-22Δ4::ckr-1(OE)*: n=11 (orange triangles). (**C**) Representative body curvature kymographs for worm locomotion during basal locomotion for indicated genotypes. Head to tail orientation along the horizontal axis in each kymograph is left to right as indicated for wild-type. Time is indicated along the vertical axis from 0 min to 1 min. (**D**) Top, representative fluorescent images of SMD motor neuron in *ckr-1(OE)* animals without (left) or with (right) miniSOG expression 16 hr following photoactivation. Bottom, representative 30 s track for control *ckr-1(OE)* (−miniSOG, left) animal or SMD ablated *ckr-1(OE)* (+miniSOG, right) animal 16 hr after photostimulation. Scale bar, 1 μm. (**E**) Average body bending angle distribution (mean ± SEM) for control *ckr-1(OE)* (green circles, n=11) and SMD ablated *ckr-1(OE)* (brown squares, n=11) animals. SMD ablation reduces the frequency of large bending angles produced by *ckr-1(OE)*. Kolmogorov-Smirnov test: \*p<0.05. (**F**) Comparison of average body bending amplitude for control *ckr-1(OE)* (n=11) and SMD ablated *ckr-1(OE)* (n=11). SMD ablation significantly reduces the enhanced body bending amplitude observed by *ckr-1(OE)*. Bars represent mean ± SEM. \*\*\*p<0.001, Student's t-test.

The online version of this article includes the following figure supplement(s) for figure 6:

**Source data 1.** Source data for frequency of bending angles (*Figure 6B*).

**Source data 2.** Source data for frequency of bending angles (*Figure 6E*).

**Source data 3.** Source data for bending amplitude (*Figure 6F*).

**Figure supplement 1.** Effects of *ckr-1(OE)* are dependent on NLP-12 and *miniSOG* expression alone does not alter SMD morphology or behavior.

**Figure supplement 1—source data 1.** Source data for frequency of bending angles (*Figure 6—figure supplement 1B*).

**Figure supplement 1—source data 2.** Source data for frequency of bending angles (*Figure 6—figure supplement 1C*).

Following the onset of photostimulation, Chrimson-expressing animals rapidly increased reorientations (*Figure 7B–C*, *Video 6*), while control animals (-Retinal) did not increase trajectory changes during the light stimulation period (*Figure 7C*). SMD photostimulation also elicited a modest increase in body bending (*Figure 7—figure supplement 1B*). Conversely, transient and inducible silencing of the SMDs by histamine-gated chloride channel expression significantly reduced reorientations during food searching (*Figure 7D*). Thus, direct activation or inhibition of SMD neurons alter turning and reorientations, consistent with a potential mechanism for NLP-12/CKR-1 modulation of local searching through signaling onto the SMD neurons.

To explore the dynamics of SMD neuronal activity during searching, we next measured combined calcium responses from SMD neurons of behaving animals. We simultaneously recorded GCaMP6s and mCherry fluorescence (*flp-22Δ* promoter) during ARS (0–5 min off food) and dispersal (30–35 min off food) (*Video 7*). We observed a striking elevation of wild-type SMD activity during ARS compared with dispersal (*Figure 8A, B, D and E*, *Figure 8—figure supplement 1*). Though overall calcium levels during ARS were positively correlated with reorientation frequency (*Figure 8D*, Pearson's correlation r=0.54), discrete events where the peak fluorescence ratio was elevated were not well correlated with specific episodes of behavior. This would be predicted for our measurements of combined fluorescence from SMDD and SMDV neurons that themselves have distinct patterns of activation (*Kaplan et al., 2020*). By comparison, SMD activity of *ckr-1(lf)* animals remained low throughout the ARS period (*Figure 8C–E*), supporting a model (*Figure 9*) where NLP-12/CKR-1 signaling promotes local searching by biasing SMD head motor neurons toward increased activation.

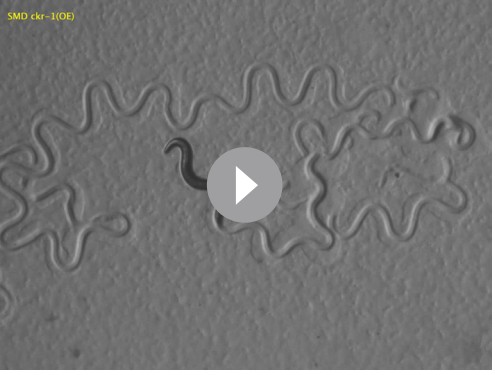

**Video 5.** Representative 20-s video showing locomotion on food of animal overexpressing *ckr-1* in the SMD motor neurons. Video has been sped up 4×.
https://elifesciences.org/articles/71747/figures#video5

## Discussion

Neuropeptidergic systems have crucial roles in modulating neuronal function to shape alternate behavioral responses, but we have limited knowledge of the circuit-level mechanisms by which these alternate responses are generated. Here, we show that the *C. elegans* NLP-12 neuropeptide system, closely related to the CCK system in mammals, shapes adaptive behavior through modulation of motor circuits dedicated to control of either head or body wall musculature. We

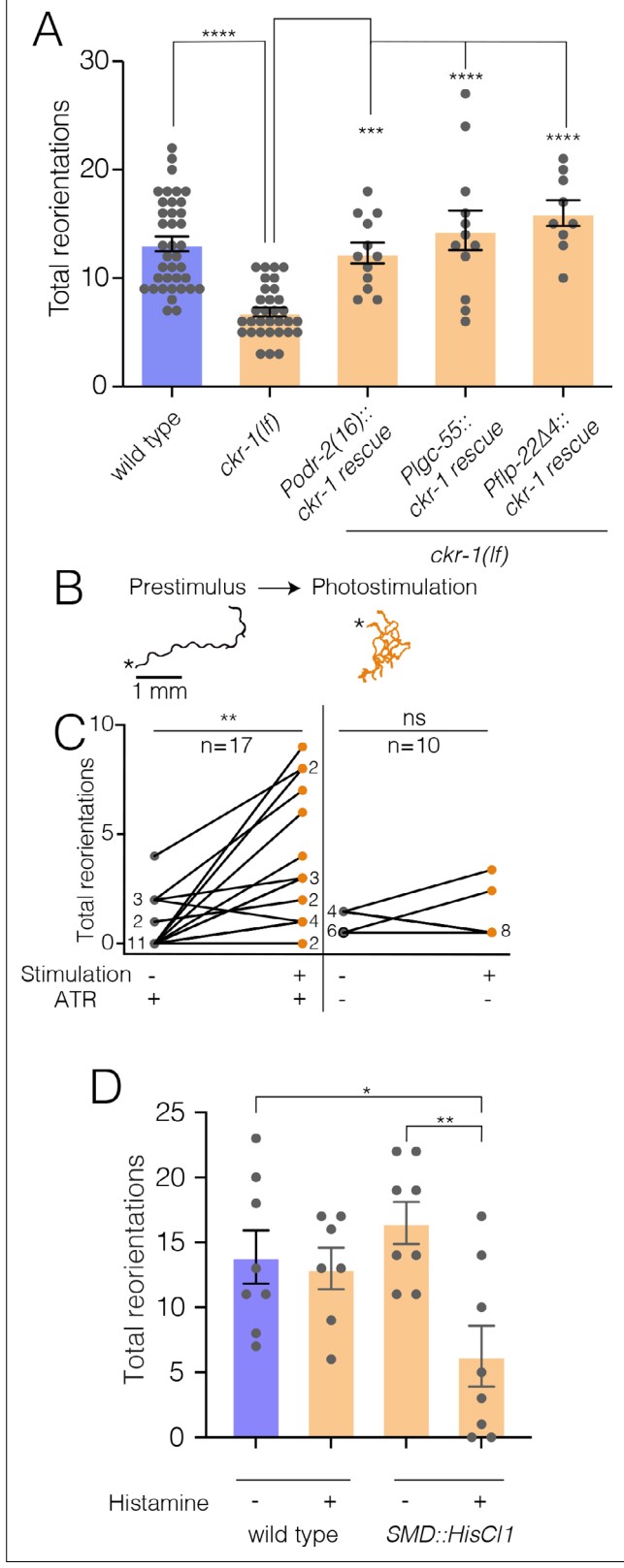

**Figure 7.** NLP-12/CKR-1 excitation of the SMD neurons promotes reorientations. Total reorientations measured during 0–5 min following removal from food for the genotypes indicated. *ckr-1* rescue refers to expression of wild-type *ckr-1* (5 ng/µl) in *ckr-1(ok2502)* animals using the indicated promoters. Bars represent mean ± SEM. ****p<0.0001, ***p<0.001 ANOVA with Holms-Sidak post hoc test. wild-type: n=38, *ckr-1(lf)*: n=32, *Podr-2(16)::ckr-1*

*Figure 7 continued on next page*

*Figure 7 continued*

*rescue*: n=12, *Plgc-55::ckr-1 rescue*: n=12, *Pflp-22(Δ4)::ckr-1 rescue*: n=9. (**B**) Representative tracks (1 min) on thinly seeded NGM agar plates prior to (left) and during photostimulation (right) for transgenic animals expressing *Podr-2(16)::Chrimson*. Scale bar, 1 mm. Asterisks (*) indicate the position of worm at the start of recording. (**C**) Left, quantification of reorientations for individual animals over 1 min durations prior to (prestimulus) and during photostimulation (+ATR). Right, quantification of reorientations for individual animals prior to and during photostimulation in control animals (−ATR). Black circles, reorientations during prestimulus. Orange circles, reorientations during photostimulation. Numbers adjacent to circles indicate number of overlapping data points. **p<0.01, ns, not significant. Paired t-test. ATR, all-trans retinal. (**D**) Quantification of reorientations for wild-type and transgenic animals, (*Pflp-22Δ4::His-Cl1::SL2::GFP*), in the presence and absence of histamine. Note reduced reorientations with SMD silencing in transgenics (+histamine). **p<0.01, *p<0.05, ANOVA with Holms-Sidak post hoc test. wild-type: −Histamine: n=8, +Histamine: n=7, *pSMD::HisCl1::SL2::GFP*: −Histamine: n=8, +Histamine: n=8. NGM, nematode growth media.

The online version of this article includes the following figure supplement(s) for figure 7:

**Source data 1.** Source data for reorientations quantified during area restricted search (0–5 min off food, *Figure 7A*).

**Source data 2.** Source data for reorientations quantified during SMD photostimulation (*Figure 7C*).

**Source data 3.** Source data for reorientations quantified during area restricted search upon SMD silencing (0–5 min off food, *Figure 7D*).

**Figure supplement 1.** SMD activation modestly impacts body bending.

**Figure supplement 1—source data 1.** Source data for frequency of bending angles (*Figure 7—figure supplement 1A*).

**Figure supplement 1—source data 2.** Source data for body bending amplitude quantified during SMD photostimulation (*Figure 7—figure supplement 1B*).

demonstrate that NLP-12 modulation of these circuits occurs through distinct GPCRs, CKR-1 and CKR-2, that primarily act on either head or body wall motor neurons, respectively. Under basal conditions, we suggest that NLP-12 modulation of the body wall motor circuit predominates, influencing the depth of body bends during sinusoidal movement through CKR-1 and CKR-2 GPCRs located on body wall motor neurons. NLP-12 activation of head motor neurons through CKR-1 becomes predominant in the absence of food, promoting reorientations. We propose that changes in food availability reconfigure functional connectivity in the NLP-12 system by differentially engaging GPCRs across the head and body wall motor circuits. Intriguingly, the involvement of two GPCRs in nematode NLP-12 signaling is reminiscent of the organization of the CCK system in rodents, which relies on signaling through CCK1 and CCK2 GPCRs (*Janssen et al., 2009*). New details about central CCK signaling and the brain GPCRs involved are continuing to emerge (*Ballaz, 2017*; *Chen et al., 2019*; *Crosby et al., 2018*; *Lee and Soltesz, 2011*; *Li et al., 2014*; *Miyasaka and Funakoshi, 2003*; *Nishimura et al., 2015*; *Saito et al., 1980*). Our findings may point toward similar utilization of specific CCK-responsive GPCRs to coordinate activity across mammalian brain circuits.

NLP-12 neuropeptides act as key modulators in a range of *C. elegans* behaviors. Local search responses to varying oxygen levels and decreased

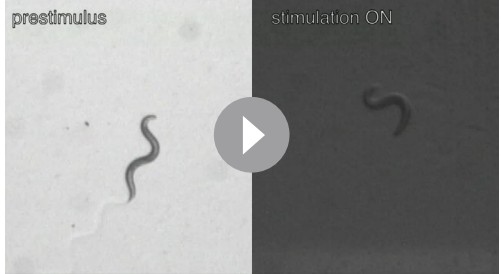

**Video 6.** Representative 20-s video showing locomotion on food of animal in the absence (left) and during SMD photostimulation (right). Video has been sped up 4×.

https://elifesciences.org/articles/71747/figures#video6

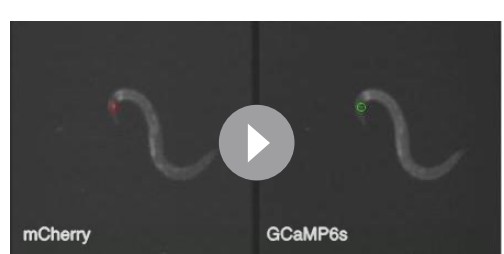

**Video 7.** Representative 20-s video showing simultaneous post hoc tracking of mCherry and GCaMP6s fluorescence for ratiometric calcium imaging analysis. Video has been sped up 4×.

https://elifesciences.org/articles/71747/figures#video7

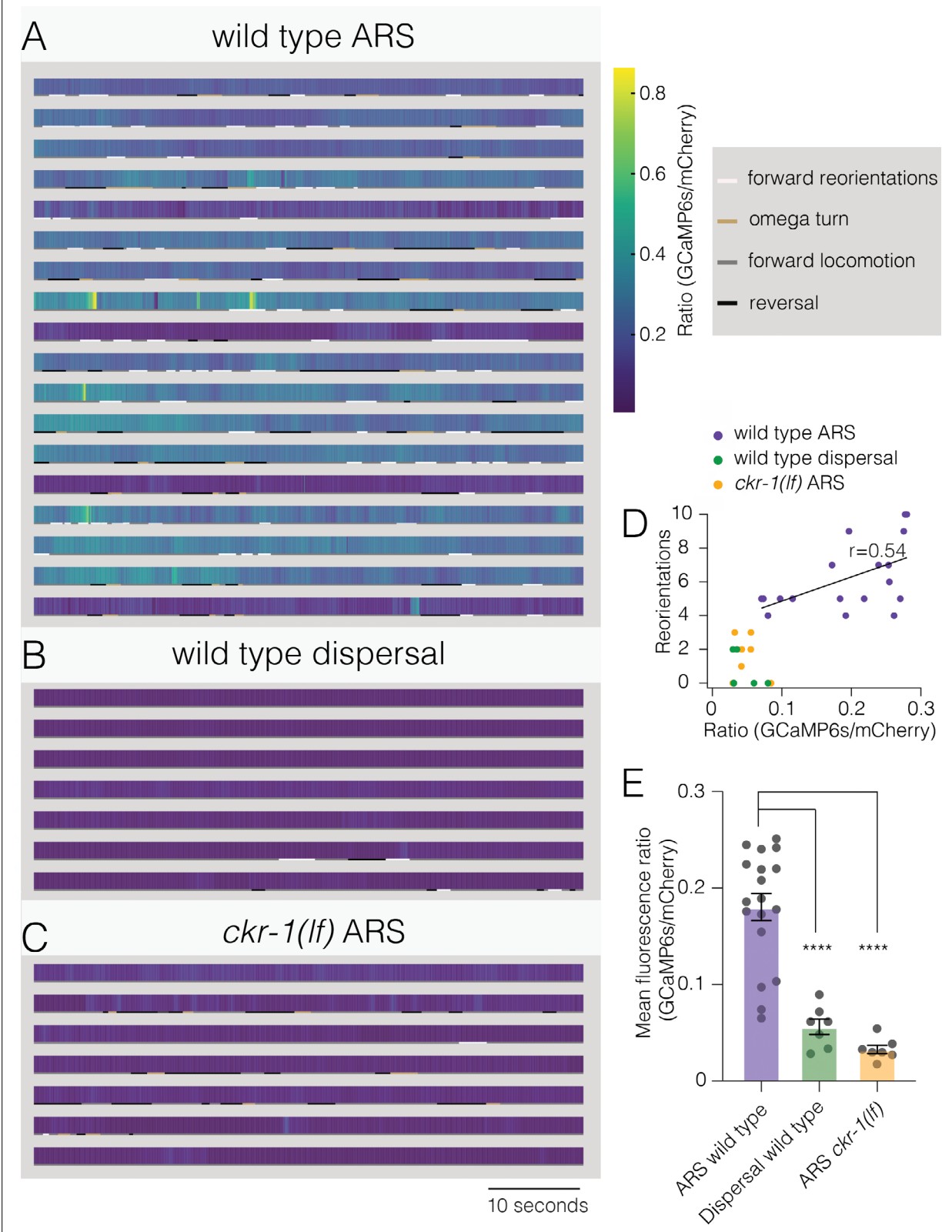

**Figure 8.** Elevated activity in SMD motor neurons during ARS promotes reorientations. (**A–C**) Representative heat maps showing activity of SMD neurons in transgenic animals (*Pflp-22Δ4::GCaMP6s::SL2::mCherry*) during ARS (**A**) and dispersal (**B**) for wild type, and ARS for *ckr-1(ok2502)* (**C**). Each row represents one animal over a duration of 1 min. Corresponding behaviors (forward, reversal, omega turn, forward reorientation) are annotated by color-coded (as indicated in legend) horizontal bar below each heat map. The SMD GCaMP6s/mCherry fluorescence ratio is elevated during wild-type

*Figure 8 continued on next page*

*Figure 8 continued*

ARS, compared with either *ckr-1(lf)* ARS, and wild-type dispersal. (**D**) Number of reorientations plotted against mean SMD GCaMP6s/mCherry ratio for the individuals in (**A–C**). Black line indicates linear fit for wild-type ARS values, with Pearson's correlation coefficient (r), *p=0.02. (**E**) Quantification of mean SMD fluorescence ratio (GCaMP6s/mCherry) during ARS or dispersal for the genotypes indicated. ****p<0.0001, ANOVA with Holms-Sidak post hoc test. ARS wild-type: n=18, ARS *ckr-1(ok2502)*: n=7, Dispersal wild-type: n=7. ARS, area-restricted searching.

The online version of this article includes the following figure supplement(s) for figure 8:

**Source data 1.** Source data for GCaMP6s/mCherry ratio during SMD calcium imaging (*Figure 8A–D*).

**Source data 2.** Source data for mean GCaMP6s/mCherry ratio during SMD calcium imaging (*Figure 8E*).

**Figure supplement 1.** Representative calcium signals (GCaMP6s/mCherry ratio) for wild-type ARS, wild-type dispersal, and *ck-1(lf)* ARS.

food availability both involve NLP-12 signaling (*Bhattacharya et al., 2014*; *Hums et al., 2016*). Additionally, NLP-12 signaling has been implicated in various aspects of proprioceptive signaling and postural control (*Hu et al., 2015*; *Hu et al., 2011*). However, the mechanisms by which NLP-12 peptides exert their influence over these diverse behavioral responses have remained unclear. Our work addresses these mechanistic questions by defining roles for CKR-1 and CKR-2 GPCRs during basal locomotion and ARS. ARS is a complex motor behavior, involving rapid trajectory changes that serve to maintain the animal within a restricted area of their immediate environment (*Bhattacharya et al., 2014*; *Calhoun et al., 2014*; *Gray et al., 2005*; *Hums et al., 2016*). Reorientations during searching are produced through high angle forward turns (*Bhattacharya et al., 2014*; *Broekmans et al., 2016*; *Pierce-Shimomura et al., 1999*) and reversal-coupled omega turns (*Bhattacharya et al.,*

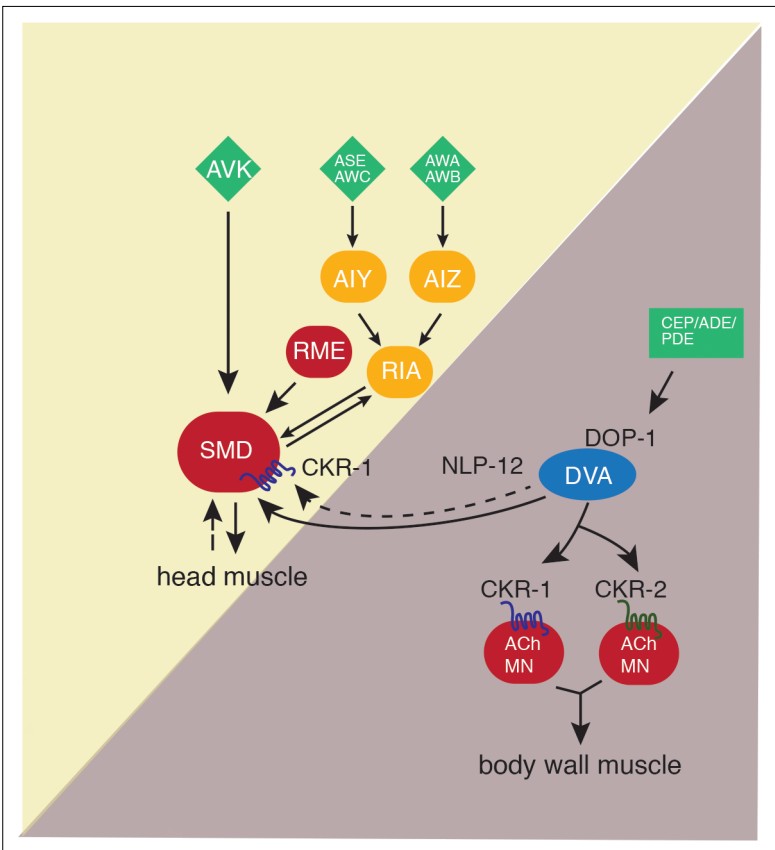

**Figure 9.** Proposed model for NLP-12 action through CKR-1 and CKR-2. During basal locomotion, NLP-12 activation of CKR-1 and CKR2 GPCRs in ventral nerve cord motor neurons regulates body bending. During local searching, NLP-12 acts primarily through CKR-1 in SMD motor neurons to promote increased turning, trajectory changes and enhance body bending. Solid arrows indicate known synaptic connections, dotted arrows indicate extrasynaptic. Sensory neurons (green), head interneurons (orange), and motor neurons (red). Olfactory sensory neurons: AWA, AWB, AWC, and ASE.

2014; *Gray et al., 2005*). We previously demonstrated a requirement for NLP-12 in promoting reorientations during local searching. (*Bhattacharya et al., 2014*). Our analysis here shows that loss of *nlp-12* also has modest effects on body posture during normal exploratory movement, indicating NLP-12 regulation of motor targets under basal conditions. Intriguingly, the behavioral requirement for NLP-12 is far more apparent during local searching compared with basal locomotion, suggesting enhanced involvement of NLP-12 signaling for performance of local searching. Similar observations about NLP-12 involvement in chemotactic responses to varying oxygen levels suggested a model for graded NLP-12 regulation of movement (*Hums et al., 2016*). Based on our observations, we speculate that increased engagement of head motor neurons through CKR-1 activation may be a generalizable mechanism for dynamic NLP-12 regulation of behavior over changing external conditions.

Prior studies had implicated the CKR-2 GPCR in NLP-12 function (*Hu et al., 2015*; *Hu et al., 2011*; *Janssen et al., 2008*), but roles for CKR-1 had not been previously described. Our genetic analyses and heterologous expression studies firmly establish CKR-1 as a functional target for NLP-12 signaling with an activation profile similar to CKR-2. CKR-2 shows slightly broader expression compared with CKR-1, but both GPCRs are expressed across a variety of neuron classes, including many that do not receive direct synaptic inputs from DVA. We noted very little overlap in CKR-1 and CKR-2 expression, consistent with the idea that the two GPCRs serve distinct roles in modulating behavior. NLP-12 activation of CKR-2 stimulates neurotransmission through coupling with *egl-30* ($G_{\alpha q}$) and *egl-8* (PLCβ) likely by DAG interaction with the synaptic vesicle priming factor UNC-13 (*Hu et al., 2015*; *Hu et al., 2011*). Given the sequence homology between CKR-1 and CKR-2, it seems likely that CKR-1 also functions to positively regulate neuronal activity through *egl-30*. In support of this idea, we found that SMD-specific CKR-1 overexpression and SMD neuron photostimulation produced qualitatively similar behavioral effects. The DVA neuron makes a single synapse with SMDVL (Worm wiring). While it is possible that this single synapse accounts for NLP-12 elicited behavioral changes during local searching, it seems likely that extrasynaptic signaling to other SMD neurons also contributes.

Prior studies have indicated SMDs are cholinergic and their stimulation is sufficient to produce $Ca^{2+}$ transients in head/neck muscles, consistent with proposed roles in head bending (*Pereira et al., 2015*; *Shen et al., 2016*). Prior studies of worms immobilized using microfluidic chips and freely moving animals noted anti-phasic activity between SMDD and SMDV neurons and opposing head/neck musculature during head bending (or head casting) (*Hendricks et al., 2012*; *Kaplan et al., 2020*; *Shen et al., 2016*; *Yeon et al., 2018*). Our $Ca^{2+}$ imaging studies did not offer sufficient cellular resolution to directly address this point. However, combined with our silencing, photostimulation and CKR-1 overexpression experiments, our SMD $Ca^{2+}$ imaging provides strong evidence that NLP-12 activation of CKR-1 modulates functional connectivity between SMD neurons and their partners. Physiological regulation of SMD activity is complex and involves reciprocal connections with RIA interneurons, reciprocal signaling with RME motor neurons, as well as proprioceptive feedback (*Hendricks et al., 2012*; *Ouellette et al., 2018*; *Shen et al., 2016*; *White, 2018*; *White et al., 1997*; *Yeon et al., 2018*). In particular, inhibitory signaling from the GABAergic RME neurons onto the SMDs is implicated in modulation of head bending amplitude to optimize head bends for forward movement. While the precise role of NLP-12 modulation of SMD activity remains unclear, one intriguing possibility is that NLP-12-elicited increases in SMD activity uncouple the SMDs from RME inhibitory regulation, perhaps promoting large amplitude head swings that couple to forward reorientations during searching. We propose that elevated SMD activity is permissive for reorientations to occur, perhaps acting in concert with SMD proprioceptive functions (*Yeon et al., 2018*) or other neurons implicated in the regulation of head movement and turning, such as SMB (*Oranth et al., 2018*).

Surprisingly, selective *ckr-1* overexpression using the *odr-2(16)* or *flp-22Δ4* promoters increased body bend depth, raising the question of how altered SMD activity might translate into increased body bending. Recent work suggests an interesting functional coupling between the activity of SMD neurons and ventral cord B-type motor neurons (*Kaplan et al., 2020*). B-type motor neurons are suggested to act as a distributed central pattern generator for the propagation of body bends (*Gao et al., 2018*; *Xu et al., 2018*). CKR-1 activation of SMDs may therefore influence body depth directly by altering body wall motor neuron excitability through a gap junction connection between VB1 and SMDVR or through neuromuscular synapses located in the sub-lateral processes.

The similar potency of NLP-12 peptides for activating CKR-1 and CKR-2, suggests that differential contributions of these GPCRs during basal locomotion and search responses do not arise due to

dramatic differences in NLP-12 potency to activate each receptor. This raises important questions about how a bias toward CKR-1 modulation of the head motor circuit during local searching may occur. We envision that NLP-12 regulation of the SMD neurons acts in parallel with other neural pathways previously shown to promote reversals during local searching. For example, olfactory information about food availability is conveyed by sensory neurons such as AWC and ASK to premotor interneurons (AIA, AIB, AIY) and ultimately transformed into patterns of motor neuron activity that drive reversals (*Gray et al., 2005*; *Hills et al., 2004*; *Ouellette et al., 2018*; *Sawin et al., 2000*). The SMD neurons also receive synaptic information from this circuit (e.g., through synaptic connections from the AIB and RIM neurons) (*White et al., 1997*), raising the possibility that a pathway activated by food removal may enhance SMD sensitivity to CKR-1 activation. In this case, SMD neurons may be a site for integration of information encoding reversals and forward reorientations during local searching. A shift to CKR-1 modulation of head neurons

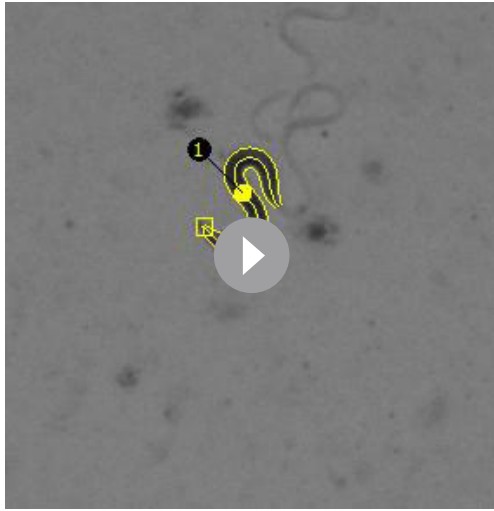

**Video 8.** Representative 20-s video showing tracking locomotion of animal overexpressing *nlp-12* in WormLab to analyze body bending. Video has been sped up 4×.
https://elifesciences.org/articles/71747/figures#video8

during searching could also be triggered by dopaminergic stimulation of DVA. Prior work implicated dopaminergic signaling from PDE neurons in the regulation of NLP-12 and motor responses (*Bhattacharya et al., 2014*; *Oranth et al., 2018*). In this case, elevated levels of NLP-12 secretion, perhaps from release sites in the nerve ring region, would be predicted to bias the system toward enhanced activation of the SMD neurons and elicit increased turning. Notably, PDE also regulates an antagonistic peptidergic circuit, mediated by FLP-1 neuropeptides, through inhibitory connections with AVK interneurons (*Oranth et al., 2018*), suggesting potentially more distributed behavioral regulation.

Our studies of the nematode NLP-12 system offer new mechanistic insights into neuropeptide modulation of behavior. Our findings provide a key first step in defining roles for two NLP-12-responsive GPCRs in coordinating motor control across changing conditions. We propose that the NLP-12 system conditionally engages GPCRs expressed in head or body motor neurons to modify specific features of locomotion, most notably reorientations during searching and body bend depth during basal locomotion. Brain CCK has been increasingly implicated as a key regulator in diverse aspects of behavior, including feeding, satiety, memory, nociception, and anxiety (*Ballaz, 2017*; *Chandra and Liddle, 2007*; *Liddle, 1997*; *Miyasaka and Funakoshi, 2003*; *Lajtha and Lim, 2006*; *Rehfeld, 2017*). Thus our studies elucidating mechanisms for NLP-12 regulation of circuit function in the compact nematode nervous system may have important and broadly applicable implications for neuromodulation in more complex systems.

## Materials and methods
### Strains
All nematode strains (*Supplementary file 1*) were maintained on OP50 seeded agar nematode growth media (NGM) at room temperature (22–24°C). N2 Bristol strain was used as wild type. Transgenic animals were generated by microinjection into the germ line and transformation was monitored by co-injection markers. Multiple independent extrachromosomal lines were obtained for each transgenic strain and data were presented from a single representative transgenic line. Stably integrated lines were generated by X-ray integration and outcrossed at least four times to wild type.

## Molecular biology

All plasmids, unless specified, were generated by Gateway cloning (see *Supplementary files 1–5*). p-ENTR plasmids were generated for all promoters used (*Supplementary file 5*). The *ckr-1* minigene construct (pRB12/pRB13) was generated by cloning the *ckr-1* coding sequence (start to stop), with introns 1, 8, and 9. For cell-specific overexpression or rescue, the *ckr-1* minigene was recombined with entry vectors containing the relevant cell-specific promoters (*Supplementary files 3-4*).

## Behavioral assays and analyses

All behavioral assays were carried out using staged 1 day adult animals on Bacto-agar NGM agar plates seeded with a thin lawn of OP50 bacteria (50 µl) unless otherwise noted. Video recordings for behavioral analyses were obtained using a

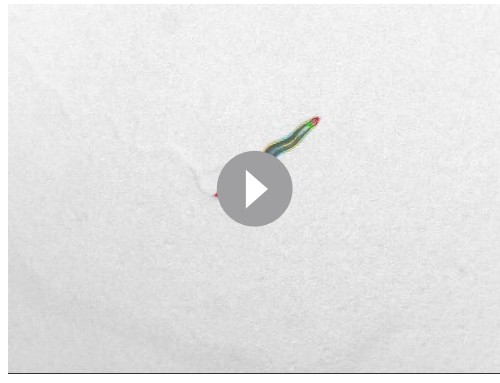

**Video 9.** Representative 20-s video showing single worm tracking of wild-type animal during basal locomotion on food to analyze body bending and head bending. Video has been sped up 4×.
https://elifesciences.org/articles/71747/figures#video9

Firewire camera (Imaging Source) and ICCapture2.2. Animals were allowed to acclimate for 30 s prior to video recording. Post hoc locomotor analysis was performed using WormLab (MBF Bioscience) (*Video 8*). Videos were thresholded to detect worms, and worm movement was tracked. Body bend amplitude was quantified as the average centroid displacement over the duration of a locomotion track (*Figure 1B*). Body bending angle was measured, at the midbody vertex, as the supplement of the angle between the head, mid-body, and tail vertices (*Figure 1C*). Bending angles were measured, continuously for each frame tracked, over 30 s (900 frames @30 fps). The measured bending angles were binned to generate a frequency distribution of body bending angles. Kymographs were generated from worm body curvature data (WormLab) in MATLAB (MathWorks, Natick, MA).

## Area restricted search behavior

For quantification of local search behavior, single well-fed animals were transferred to an intermediate unseeded plate. After 1 min, animals were repicked without bacteria and transferred to an unseeded behavior assay plate. Digital movies were captured over the first 5 min (local search) and after 30 min (dispersal) following removal from food. Reorientations were manually scored post hoc from monitoring movement direction, over sequential frames (~200 frames for forward reorientations, ~ 600 frames for reversal-coupled omega turns) from the start of the reorientation (original trajectory) to when the animal completed the reorientation (new trajectory) (*Figure 3B*, *Figure 3—figure supplement 1*). A forward reorientation was scored after animals moved a minimum of 3 s (~100 frames @30 fps) along a new trajectory. We scored forward trajectory changes >50° and reversal coupled omega turns as reorientations (examples of each in *Figure 3B*, *Figure 3—figure supplement 1*). Trajectory changes where animals initially performed head bends >50°, but then resumed the original path of movement or altered immediate trajectory <50° were not scored as reorientations. Trajectory changes were quantified (in degrees) using the angle tool (ImageJ, National Institutes of Health) to measure the angle between the original and new trajectory (*Figure 3B*, *Figure 3—figure supplement 1*). We excluded reversals and post reversal changes in trajectory that did not involve omega turns.

## Single worm tracking

Single worm tracking was carried out using Worm Tracker 2 (*Yemini et al., 2011*). Animals were allowed to acclimate for 30 s prior to tracking. Movement features were extracted from 5 min of continuous locomotion tracking (*Video 9*). Worm tracker software version 2.0.3.1, created by Eviatar Yemini and Tadas Jucikas (Schafer lab, MRC, Cambridge, UK), was used to analyze movement (*Yemini et al., 2013*). Worms were segmented into head, neck, midbody, hips, and tail. The body bend angle is angle measured at the midbody vertex, between the neck and hip skeleton vertices (*Figure 2A*). Head bend angles were measured as the largest bend angle prior to returning to a straight, unbent position (*Figure 2B*). Absolute midbody bending (*Figure 2A*) and head bending (*Figure 2B*) angles

were quantified. Single worm tracking affords higher resolution and allows for rich quantification of relatively subtle postural changes. However, the continuous tracking of animals was difficult to achieve using this approach during the numerous steep turns performed during ARS, or with NLP-12 or CKR-1 overexpression. Post hoc analysis of videos to measure body bending (as described above) proved most reliable.

## SMD ablation

Conditions for cell ablation by miniSOG activation were adapted from *Xu and Chisholm, 2016*. MiniSOG activation was achieved by stimulation with repetitive 2 Hz 250 ms blue light pulses for 12 min (200 mW/cm$^2$, 488 nm 50 W LED [Mightex Systems]). Experiments were performed on unseeded plates using larval stage four *ckr-1(OE)* animals expressing miniSOG and GFP transgenes under the *flp-22Δ4* promoter. Following stimulation, animals were allowed to recover in the dark on NGM OP50 plates for 16 hr prior to behavioral analysis or imaging.

## Photostimulation experiments

All-trans retinal (ATR) plates were prepared (100 mM stock in ethanol, final working 2.7 mM in OP50). Plates were stored at 4°C under dark conditions and used within 1 week. Animals were grown on +ATR OP50 plates in dark and L4 animals were transferred to a fresh +ATR plate prior to the day of experiment. Experiments were performed using 1-day adults. For ChR2 photostimulation, experiments were conducted using a fluorescent dissecting microscope (Zeiss stereo Discovery.V12) equipped with a GFP filter set. Behavior was recorded for a 1-min period prior to photostimulation and during a subsequent 1 min period during photostimulation. Data are expressed as % change in reorientations across these time intervals. Chrimson photostimulation (26 mW/cm$^2$) experiments were conducted using a 625 nm 50 W LED (Mightex Systems). Animals were video recorded for 1 min in the absence of light stimulation (prestimulus) and subsequently for 1 min with light stimulation. Control experiments (−ATR) were performed in the same manner.

## SMD silencing

ARS assays were performed on unseeded Histamine (10 mM) and control Bacto-agar NGM plates using staged 1-day adults. For SMD silencing, transgenic animals were placed on Histamine plates, seeded with 100 µl OP-50, for 1 hr prior to experiment. ARS was quantified as described previously.

## Imaging

Fluorescent images were acquired using either BX51WI (Olympus) or Yokogawa (PerkinElmer) spinning disc confocal microscopes. Data acquisition was performed using Volocity software. Staged 1-day adult animals were immobilized using 0.3 M sodium azide on 2% agarose pads. Images were analyzed using ImageJ software.

## SMD calcium imaging

Calcium imaging was performed in behaving transgenic animals, expressing GCaMP6s::SL2::mCherry under *flp-22Δ4* promoter, on 5% agarose pads on a glass slide. Animals were treated as described for ARS and dispersal assays. Animals were tracked and videos captured, with continuous and simultaneous dual-channel (GCaMP6s and mCherry) fluorescence monitoring (*Video 7*), in the time windows of ARS (0–5 min) and dispersal (30–35 min off food). Imaging was carried out on an Axio Observer A1 inverted microscope (Zeiss) connected to a Sola SE Light Engine (Lumencor) with an Olympus 2.5× air objective, and a Hamamatsu Orca-Flash 4.0 sCMOS camera. Simultaneous GCaMP and mCherry acquisition were achieved using the optical splitter Optisplit-II (Cairn Research) with filters ET525/50M and ET632/60M, and dichroic T560Iprx-UF2 (Chroma). Image acquisition was performed using Micro-manager, at 66 ms exposure (approximately 15 fps).

ROIs encompassing cell bodies in the nerve ring, labeled by mCherry, were tracked post hoc using MATLAB (Neuron Activity Analysis, Mei Zhen, *Video 7*). Frames where tracking issues were encountered due to stage movement were excluded from analysis. The background subtracted calcium signals were plotted as a ratio (GCaMP6s/mCherry). We encoded corresponding behavior into four categories: forward locomotion, reversals, forward reorientations, and omega turns. Wild-type animals that did not perform searching (<4 reorientations during ARS) were excluded from the analysis. Correlation

analysis, including linear fits and calculation of Pearson's coefficient, was performed in Graphpad Prism. For display, heat maps were plotted in Graphpad Prism (*Figure 8*) and representative traces (*Figure 8—figure supplement 1*) were interpolated with a smoothing spline in Igor Pro (Wavemetrics, Portland, OR).

### in vitro GPCR characterization

The GPCR activation assay was performed as previously described (*Caers et al., 2014*; *Peymen et al., 2019*; *Van Sinay et al., 2017*). Briefly, CHO-K1 cells stably expressing the luminescent $Ca^{2+}$ indicator aequorin and the promiscuous $G_{\alpha 16}$ protein (ES-000-A24 cell line, PerkinElmer) were transiently transfected with *ckr-1*/pcDNA3.1, *ckr-2*/pcDNA3.1, or empty pcDNA3.1 vector. Cells were transfected with Lipofectamine LTX and Plus reagent (Invitrogen) at 60–80% confluency and grown overnight at 37°C. After 24 hr, they were shifted to 28°C overnight. On the day of the assay, transfected cells were collected in bovine serum albumin (BSA) medium (DMEM/F12 without phenol red with L-glutamine and 15 mM HEPES, Gibco, supplemented with 0.1% BSA), at a density of 5 million cells per ml, and loaded with 5 µM coelenterazine h (Invitrogen) for 4 hr at room temperature. Compound plates containing synthetic peptides in DMEM/BSA were placed in a MicroBeta LumiJet luminometer (PerkinElmer). After loading, the transfected cells were added at a density of 25,000 cells/well, and luminescence was measured for 30 s at a wavelength of 469 nm. After 30 s, 0.1% triton X-100 (Merck) was added to lyse the cells, resulting in a maximal $Ca^{2+}$ response that was measured for 30 s. To constitute concentration-response curves of NLP-12 peptides, peptide concentrations ranging from 1 pM to 10 µM were tested in triplicate on 2 independent days.

## Acknowledgements

The authors thank the *Caenorhabditis* Genetics Center, which is funded by the National Institutes of Health National Center for Research Resources, and the Mitani laboratory (National Bioresource Project) for providing *Caenorhabditis elegans* strains. The authors thank Mei Zhen lab for MATLAB script for calcium imaging analysis, Claire Bénard for strains, Michael Gorczyca and William Joyce for technical support. The author thank Francis lab members for helpful comments on the manuscript.

## Additional information

### Funding

| Funder | Grant reference number | Author |
| --- | --- | --- |
| National Institutes of Health | R21NS093492 | Michael M Francis |
| European Research Council | 340318 | Isabel Beets |
| Research Foundation Flanders | G0C0618N | Isabel Beets |

The funders had no role in study design, data collection and interpretation, or the decision to submit the work for publication.

### Author contributions

Shankar Ramachandran, Conceptualization, Data curation, Formal analysis, Investigation, Methodology, Validation, Visualization, Writing - original draft, Writing - review and editing; Navonil Banerjee, Conceptualization, Data curation, Formal analysis, Investigation, Methodology, Validation; Raja Bhattacharya, Conceptualization, Data curation, Formal analysis, Investigation, Methodology; Michele L Lemons, Data curation, Resources; Jeremy Florman, Data curation, Formal analysis, Software; Christopher M Lambert, Denis Touroutine, Kellianne Alexander, Mark J Alkema, Resources; Liliane Schoofs, Supervision; Isabel Beets, Data curation, Formal analysis, Funding acquisition, Methodology; Michael M Francis, Conceptualization, Funding acquisition, Project administration, Supervision, Writing - original draft, Writing - review and editing

Author ORCIDs
Shankar Ramachandran (iD) http://orcid.org/0000-0002-1299-4482
Michele L Lemons (iD) http://orcid.org/0000-0001-8459-4130
Jeremy Florman (iD) http://orcid.org/0000-0001-7578-3511
Mark J Alkema (iD) http://orcid.org/0000-0002-1311-5179
Michael M Francis (iD) http://orcid.org/0000-0002-8076-6668

### Decision letter and Author response
Decision letter https://doi.org/10.7554/eLife.71747.sa1
Author response https://doi.org/10.7554/eLife.71747.sa2

## Additional files

### Supplementary files
• Supplementary file 1. Stains generated/used in this work.

• Supplementary file 2. Identification (method of ID, marker and strain indicated for each neuron) to determine *ckr-1* expressing neurons. * Indicated strains were crossed into *ufIs141 (Pckr-1::ckr-1::SL2::GFP)* to generate strains to determine colocalization. #+ or – indicates presence or absence of *ckr-1* expression in identified neuron. * Indicated strains were crossed into *ufIs141* to generate strains to determine colocalization, #+ indicates *ckr-1* expression, - indicates absence.

• Supplementary file 3. Promoters used in *ckr-1(OE)* screen (*Figure 5C*) indicating expression pattern. **Bold indicates neurons where *ckr-1* is expressed.

• Supplementary file 4. Plasmid constructs used in cell specific *ckr-1(OE)* screen or cell-specific rescue (*Figures 5C and 7A*). For cell specific overexpression or rescue of *ckr-1*, *ckr-1* minigene was expressed under indicated promoters. Entry vectors containing promoters recombined with destination vectors pRB12 or pRB13 for cell-specific overexpression or rescue of *ckr-1*.

• Supplementary file 5. Promoter lengths and primer information for promoters used.

• Transparent reporting form

### Data availability
All data generated or analyzed during this study are included in the manuscript and supporting files.

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
