## [Editor Report]

In this work, Ramachandran and colleagues investigate how the *C. elegans* cholecystokinin-like neuropeptide (NLP-12) signaling pathway modulates animal posture during locomotion. They show that control over head- versus body-bending diverges at the level of two different NLP-12 receptors and that this fine-tuning enables the animal to reach different behavioral goals i.e., local exploration versus long-distance traveling during food search.

---

## [Decision Letter]

[Editors’ note: the authors submitted for reconsideration following the decision after peer review. What follows is the decision letter after the first round of review.]

Thank you for submitting your work entitled "A conserved neuropeptide system links head and body motor circuits to enable adaptive behavior" for consideration by *eLife*. Your article has been reviewed by 3 peer reviewers, one of whom is a member of our Board of Reviewing Editors, and the evaluation has been overseen by a Senior Editor. The reviewers have opted to remain anonymous.

Our decision has been reached after consultation between the reviewers. Based on these discussions and the individual reviews below, we regret to inform you that your work will not be considered further for publication in *eLife*.

While the reviewers find your work very interesting and acknowledge its importance in understanding the role of cholecystokinin signaling in differentially controlling aspects of locomotion behavior in *C. elegans*, we think that in its current form it lacks further mechanistic insights into how ckr-1 signaling controls SMD activity.

*Reviewer #1:*

In this manuscript Ramachandran et al. provide a *C. elegans* behavioral genetics study focused on the worm cholecystokinin-like neuropeptide-receptor system. They show that nlp-12 neuropeptides released from the DVA neuron fulfil a dual role in controlling body posture as well as head-bending mediated area restricted search (ARS). Previous work showed that DVA controls body posture via nlp-12 signaling to ckr-2 receptor in ventral cord motor neurons. Moreover, nlp-12 signaling was implicated in ARS; but the exact circuit mechanisms and targets of nlp-12 remained elusive. The present work shows in a pretty straight forward way that ckr-1 in SMD head motor neurons is the missing link. In worms, ARS is composed of quiet complex body movements including high angle turns during the worm's forward crawling state. Nlp-12 and ckr-1 mutants show reduced head bending during ARS, while overexpression leads to a stark ectopic ARS like behavior. The authors convincingly show that SMDs are the site of action for ckr-1 and implicated in ARS. They show both requirement and sufficiency of SMDs for ARS like behaviors. The regulation of ARS vs. dispersive behaviors has been extensively studied at the levels of sensory and interneurons in the worm, but how the switch is implemented at motor circuits was largely unknown. Conceptually, this is one of only few studies investigating the selective control of head versus body movements and provides some interesting insights into the underlying mechanisms; therefore, the study is definitely important and timely. But it is unclear still how upper sensory circuits transmit the switch between ARS and dispersal to the DVA-SMD circuit. Moreover, the present study does not investigate the signaling pathway of ckr-1 in SMDs and its role in controlling neuronal activity, e.g. via Ca++ imaging. As a sole behavioral genetics study, however, I find the manuscript quite complete. The experiments logically built upon another, and the paper is well written. My only major critique is that parts of the behavioral analyses are described with insufficient detail so that it is unclear to the expert how and what exact movements were quantified. This should be addressed by providing more detailed figure captions, methods sections, more supplemental figures and movies.

1) The authors should exclude (or separate) reversal states and post-reversal turns in their analyses when measuring head bending, body bending and turn events, but it is unclear if they did so.

2) Figure 1C and methods: it is unclear what defines a singular bending event as marked on the y-axis. Did the authors measure the maximum angle during each half-oscillation? If yes, this should be explained and how maxima were calculated etc. Or do the histograms represent all values from all recording frames. In the latter case, the y-axis labelling is misleading, and I suggest use "fraction of frames".

3) Figure 1C: these are averaged histograms of n=10-12 worms, but what is the average number of events per worm and in total?

4) Figure 1B-C, 2A etc.: to perform the measurements as depicted in upper panels is not really trivial, and I have the impression that the authors used their software packages in a black-box manner. What are the exact image processing steps to implement these measurements, i.e. how was vertex and sides of the angles exactly positioned? The authors should provide time-series of individual examples alongside with movies demonstrating how accurately the pipeline performs during complex ARS postures.

5) Figure 2B: the angles and body segments describing the head and head-bending angels should be unambiguously defined. The cartoon in 2B looks like they just measured nose movements.

6) Figure 3B: reorientation events are not sufficiently defined here. During ARS, worms frequently switch between forward-backward movement, perform post-reversal turns and in a continuous manner exhibit curved trajectories. From a trajectory like the red one in 3A, it is again not trivial to identify and discretize individual turning events with a start and an end and distinguish them from reversals and post reversal turns.

– The procedure needs to be explained in greater details with justification of parameter choice.

– How did the authors validate that the procedure performed well, especially during the complex ARS behaviors?

– Again, example trajectories and movies should be shown.

7) All histogram panels lack statistics, e.g. KS test or appropriate alternatives.

*Reviewer #2:*

Ramachandran et al. report the discovery of a *C. elegans* GPCR – CKR-1 – that mediates some of the effects of the cholecystokinin-like neuropeptide NLP-12 on posture and foraging behavior. The discovery of this receptor permits further study of this neuropeptide signaling system, which is conserved from worms to vertebrates. Although CKR-1 is expressed in many neurons, the authors show that its function in SMD head-motorneurons is especially important for control of posture and foraging. The manuscript's strengths include: (1) rigorous characterization of receptor-ligand interactions in vitro, using a cell-based assay for GPCR activation, and in vivo, using genetic analysis, (2) compelling data in support of a model in which NLP-12 regulates SMD neurons to control foraging, and (3) high-resolution analysis of *C. elegans* posture during foraging, which illustrates the complexity and richness of this behavior, and (4) the circuit model, i.e. a role for SMDs, is tested using a number of independent methods and clearly indicated. The manuscript does have some weaknesses. In addition to specific technical points listed below, the manuscript discussed neuropeptides derived from a single source, the DVA pre-motor neuron, acting on distinct targets via distinct receptors in a conditional manner. This interesting model is suggested by the title and the abstract and comes up plainly in the introduction and discussion. However, the model is not clearly supported by the data, which primarily focus on the characterization of CKR-1 as a relevant receptor for NLP-12 peptides. Another weakness in the manuscript arises from the authors' switching between various assays for posture during locomotion, which makes it difficult for the reader to compare data between figures. Rich kymography data are relegated to supplementary figures, and data from only a subset of relevant genotypes are shown as kymographs. The manuscript would be strengthened by more uniform analysis of posture and foraging. Finally, while the data clearly show that effects of NLP-12 on posture and foraging require SMD neurons, the manuscript does not investigate how NLP-12 affects SMD activity. The manuscript would be strengthened by experiments showing a functional connection between DVA and SMD neurons, e.g. functional imaging of SMDs during optogenetic manipulation of DVAs.

1. One premise of the work is that DVA neurons are the sole source in vivo of NLP-12 peptides. A recent study (Tao et al. 2019, Dev. Cell) shows that there is an alternate source of NLP-12, the PVD nociceptors. The authors should address the possibility that their assays also detect a contribution of PVD neurons to posture/foraging.

2. The text associated with Figure 1B-C is tentative with respect to assigning redundant functions to CKR-1 and CKR-2. Why? The data are clear; these receptors function redundantly.

3. The very nice in vitro analysis of NLP-12 receptors should include negative controls. Ideally, the authors would use a scrambled neuropeptide or a related neuropeptide to demonstrate specificity of the interactions between NLP-12 and CKR-1/2.

4. The different 'bending angles' used in Figures 1 and 2 make it difficult to compare data between figures. Also, the schematics used to explain the bending angles have small fonts and are hard to read.

5. Figure 3E shows the results of a nice experiment in which optogenetic activation of NLP-12-expressing cells – presumably DVA – causes reorientations. The authors assert that this effect requires CKR-1 but not CKR-2. The data, however, suggest that CKR-2 might have an effect. The variance of the data does not allow the authors to reject a null hypothesis, but they err in then assuming that this means that CKR-2 plays no role in the phenomenon. This experiment should be repeated to determine whether there is indeed a specific or privileged role for CKR-1 in mediating NLP-12-dependent reorientations.

6. Also, Figure 3E should show raw data – don't show proportional changes – and all Figure 3 should be scatter plots allowing the reader to assess the variance of the data.

7. The authors show that effects of receptor overexpression are suppressed by loss of NLP-12 peptides. Is there precedent for this kind of genetic interaction in the literature?

8. Also, the authors, assert that suppression of effects of CKR-1 overexpression by loss of NLP-12 shows that NLP-12 peptides are the sole ligands for this receptor (page 9, line 17). It is not clear why the authors reach this conclusion.

9. There are some very nice data that are assigned to supplementary figures but might be better placed in main figures. Figure S3A-B shows data that are integral to the authors' model and could be presented in a main figure. Also, the localization of NLP-12::Venus in DVA axons near SMD processes would be appropriate to show in a main figure. It would be ideal to mark SMDs with a red fluor so that NLP-12::Venus colocalization with SMD processes could be assessed.

10. The kymography data are nice but incomplete. The authors should show kymographs from strains of all relevant genotypes. This would include: (1) ckr-1(oe); nlp-12, (2) nlp-12, ckr-1, and ckr-2 single mutants, and (3) ckr-1; ckr-2 double mutants.

11. Page 12, last paragraph indicates that 'low levels' of expression rescue ckr-1 phenotype – how has the expression level been determined? I guess that the authors refer to the amount of DNA used for transgenesis, not a direct measure of transgene expression – this should be reworded.

12. The manuscript would be strengthened by experiments that measured the effect of DVA activation on SMD physiology and what contribution NLP-12 signaling makes to any functional connection between these neurons. One potential impact of this work is that it establishes a nice paradigm for new molecular genetic analyses of neuropeptide signaling. Direct observation of the effects of NLP-12 peptides on SMD neuron physiology would further strengthen the authors' conclusions and suggest mechanisms by which CKR-1 regulates cell physiology.

*Reviewer #3:*

In this manuscript, Ramachandran and colleagues describe how cholecystokinin-related NLP-12 neuropeptide signalling in *C. elegans* can regulate two different behavioural programmes, area-restricted search (ARS) and basal locomotion, by conditionally engaging different specific receptors that are expressed in different neuronal targets. They thoroughly characterise the CKR-1 receptor which had not been described previously, and place its function in context with that of the previously known NLP-12 receptor CKR-2. The manuscript gives new insight into an interesting and likely conserved mechanisms of how neuromodulatory systems enable adaptive behaviour by coordinating the action of neural circuits even when they are not directly connected. The conclusions drawn appear solid and are justified by the data presented, and the experimental approaches and results are well documented.

The main problem with the work is a certain lack of clarity regarding the separation of the roles of the CKR-1 and CKR-2 receptors on basal locomotion/body bending and head bending/reorientations. Overexpression of NLP-12 places animals in a chronic ARS state, as described in a previous publication. Is the NLP-12 overexpression model representative of the increased reorientation in area restricted search, or of control of undulations in basal locomotion, or both? If it is primarily representative of area restricted search, this would mean that CKR-2, similarly to CKR-1, mediates the chronic ARS state induced by NLP-12 overexpression, because in Figure 1B and C its mutation causes a reduction in the phenotype, and deletion of both ckr-1 and ckr-2 causes a stronger reduction.

Also, it is unconvincing that SMD neurons do not express ckr-2 (see S3D); no comparison of ckr-1 and ckr-2 expression levels in SMD is provided and in fact the CeNGEN data of single cell RNAseq of *C. elegans* neurons shows similar expression of both receptors in SMDD (accessible at cengen.shinyapps.io/SCeNGEA). On the other hand, loss of ckr-2 on its own does not cause a significant reduction in ARS (Figure 3A).

To clarify this, the authors could measure the reorientation rate in the nlp-12OE ckr-2 mutant strain.

Given that ckr-1 overexpression as shown in Figures4-6 increases both body bending amplitude and ARS-like high reorientation rate, the authors offer the interesting possibility that SMD may also affect basal locomotion. I would suggest an experiment that clarifies whether SMD also controls body bending in basal locomotion using the single-worm tracking assay shown in Figure 2A with the SMD-specific ckr-1 rescue strains in a ckr-1 mutant background (as used in figure 7). Also they could measure body bending in the existing data on the SMD::Chrimson optogenetics.

Overall, the manuscript is of high quality and significant interest and warrants publication in *eLife*, once those points have been addressed.

[Editors’ note: further revisions were suggested prior to acceptance, as described below.]

Thank you for resubmitting your work entitled "A conserved neuropeptide system links head and body motor circuits to enable adaptive behavior" for further consideration by *eLife*. Your revised article has been evaluated by Ronald Calabrese (Senior Editor) and a Reviewing Editor.

The reviewers are very excited about your work and the improvements made in your revision. However, there are some remaining issues that need to be addressed. As outlined below, the major concern of reviewers #1-2 with respect to SMD Ca++ imaging and the minor points of reviewer #3 should be addressed. We assume that this would not require new experiments but further in-depth analysis of the recordings, which should be all feasible in a short time frame.

*Reviewer #1:*

The authors have fully addressed my review comments to the previous submission.

In the present manuscript, the authors provide SMD Ca++ imaging experiment, which require further clarifications:

1) SMD activity in crawling worms has been reported previously by several labs, and all studies found consistently activity related to head-bending (Hendricks et al., 2012; Kaplan et al., 2020; Yeon et al., 2018). From the activity profiles shown in Figure 5, it is not possible to evaluate whether these data can be reproduced by the authors. If movie S9 is representative for these recordings, then SMD activity profiles should relate to head-bending. The fluctuations in Figure 5E rather appear as noise to me. I find it essential that the authors annotate behaviour in these recordings (forward crawling, backward crawling, head bending) and analyse the relationship between SMD activity and locomotion. If any discrepancies to the literature remain, this and possible explanations should be discussed.

2) Despite the concerns above, I find it surprising and interesting that the authors observe different perhaps baseline ratio values in 0-5min vs 30-35min off-food conditions. How this relates to the different behavioral phenotypes, incorporating our knowledge about SMD physiology remains to be discussed in more detail.

Hendricks, M., Ha, H., Maffey, N., and Zhang, Y. (2012). Compartmentalized calcium dynamics in a *C. elegans* interneuron encode head movement. Nature 487, 99-103.

Kaplan, H.S., Salazar Thula, O., Khoss, N., and Zimmer, M. (2020). Nested Neuronal Dynamics Orchestrate a Behavioral Hierarchy across Timescales. Neuron 105, 562-576.e569.

Yeon, J., Kim, J., Kim, D.-Y., Kim, H., Kim, J., Du, E.J., Kang, K., Lim, H.-H., Moon, D., and Kim, K. (2018). A sensory-motor neuron type mediates proprioceptive coordination of steering in *C. elegans* via two TRPC channels. PLoS biology 16, e2004929.

*Reviewer #2:*

The revised manuscript of Ramachandran, Francis and colleagues addresses many of the questions raised during the initial round of review. I have only one comment. The authors include new data reporting SMD activity during area-restricted search and dispersal. This is an interesting experiment that shows clear evidence for CKR-1 in regulation of SMDs. It is not clear why the authors only show data from 3 individuals in panel E of Figure 7; panel F indicates that there are many more individuals that were assayed. I suggest that the authors show data from all the individuals in panel E.

*Reviewer #3:*

Using a variety of different approaches, Ramachandran and colleagues make a convincing case that CKR-1 acting in SMD primarily mediates the effect of the NLP-12 on local food searching. All relevant weaknesses of the first submission raised by reviewers have been addressed, in particular:

– The various behavioural measurements are now well defined.

– Additional functional data have been added for SMD which strengthen the hypothesis that ckr-1-mediated signalling in SMD is both necessary and sufficient for ARS.

– New data added describing the rescue of ckr-2 expression in a ckr-1; ckr-2 double mutant is convincing to answer the question of distinguishing the (lack of a) role of ckr-2 in ARS.

– New data on SMD now answer the question whether it affects only area-restricted search or also body bending in basal locomotion.

The manuscript still a variety of different behavioural analyses, but I think this is acceptable, because they are now better defined, and are also necessary to address the different effects of overexpression/loss of function and of the different behavioural programmes controlled by NLP-12.

In the first reviews the point was made that the paper was more or less only a behavioural genetics study; I believe that the evidence from ontogenetic, chemogenetic and functional neuronal imaging approaches used to support the hypothesis, the revised study is of sufficient standard for *eLife*.

---

## [Author Response]

Reviewer #1:In this manuscript Ramachandran et al. provide a *C. elegans* behavioral genetics study focused on the worm cholecystokinin-like neuropeptide-receptor system. They show that nlp-12 neuropeptides released from the DVA neuron fulfil a dual role in controlling body posture as well as head-bending mediated area restricted search (ARS). Previous work showed that DVA controls body posture via nlp-12 signaling to ckr-2 receptor in ventral cord motor neurons. Moreover, nlp-12 signaling was implicated in ARS; but the exact circuit mechanisms and targets of nlp-12 remained elusive. The present work shows in a pretty straight forward way that ckr-1 in SMD head motor neurons is the missing link. In worms, ARS is composed of quiet complex body movements including high angle turns during the worm's forward crawling state. Nlp-12 and ckr-1 mutants show reduced head bending during ARS, while overexpression leads to a stark ectopic ARS like behavior. The authors convincingly show that SMDs are the site of action for ckr-1 and implicated in ARS. They show both requirement and sufficiency of SMDs for ARS like behaviors. The regulation of ARS vs. dispersive behaviors has been extensively studied at the levels of sensory and interneurons in the worm, but how the switch is implemented at motor circuits was largely unknown. Conceptually, this is one of only few studies investigating the selective control of head versus body movements and provides some interesting insights into the underlying mechanisms; therefore, the study is definitely important and timely. But it is unclear still how upper sensory circuits transmit the switch between ARS and dispersal to the DVA-SMD circuit. Moreover, the present study does not investigate the signaling pathway of ckr-1 in SMDs and its role in controlling neuronal activity, e.g. via Ca++ imaging.As a sole behavioral genetics study, however, I find the manuscript quite complete. The experiments logically built upon another, and the paper is well written. My only major critique is that parts of the behavioral analyses are described with insufficient detail so that it is unclear to the expert how and what exact movements were quantified. This should be addressed by providing more detailed figure captions, methods sections, more supplemental figures and movies.

We thank the reviewer for pointing this out. We have now updated the methods and figures with additional details of the experiments and their analyses. In particular, Figure 3B has been revised to show how trajectory changes were measured and reorientations scored. We also now include in Figure S3 representative sequences of frames demonstrating examples of both forward reorientations and reversal coupled omega turns from our analyses.

1) The authors should exclude (or separate) reversal states and post-reversal turns in their analyses when measuring head bending, body bending and turn events, but it is unclear if they did so.

We scored forward reorientations greater than 50 degrees and reversal-coupled omega turns as reorientations (examples of each in Figure S3). We excluded post-reversal changes in trajectory that did not involve an omega turn. We have clarified this in the Methods. We agree that it is important to know which classes of reorientation may be affected by NLP-12 and CKR-1 signaling. We now include additional experiments showing that forward reorientations are significantly reduced by deletion of either *nlp-12* or *ckr-1*, while reversal coupled omega turns are not, [Figure S4] consistent with our previously published findings for *nlp-12* (Bhattacharya *et al.,* 2015).

2) Figure 1C and methods: it is unclear what defines a singular bending event as marked on the y-axis. Did the authors measure the maximum angle during each half-oscillation? If yes, this should be explained and how maxima were calculated etc. Or do the histograms represent all values from all recording frames. In the latter case, the y-axis labelling is misleading, and I suggest use "fraction of frames".

We agree with the reviewer and clarify this in the revised version. We measured bending angles continuously (900 frames @ 30 fps) over the course of 30 s recordings. These were binned and represented as a frequency histogram. We changed the Y axis labels of the histograms to “frequency of bending angles (%)”.

3) Figure 1C: these are averaged histograms of n=10-12 worms, but what is the average number of events per worm and in total?

See above. Bending angles were measured continuously over 30 s (900 frames @ 30 fps).

4) Figure 1B-C, 2A etc.: to perform the measurements as depicted in upper panels is not really trivial, and I have the impression that the authors used their software packages in a black-box manner. What are the exact image processing steps to implement these measurements, i.e. how was vertex and sides of the angles exactly positioned? The authors should provide time-series of individual examples alongside with movies demonstrating how accurately the pipeline performs during complex ARS postures.

We now provide an illustrative time series as well as representative movies and additional details in the Methods.

5) Figure 2B: the angles and body segments describing the head and head-bending angels should be unambiguously defined. The cartoon in 2B looks like they just measured nose movements.

We have modified the figure to improve clarity and increased the size of schematics.

6) Figure 3B: reorientation events are not sufficiently defined here. During ARS, worms frequently switch between forward-backward movement, perform post-reversal turns and in a continuous manner exhibit curved trajectories. From a trajectory like the red one in 3A, it is again not trivial to identify and discretize individual turning events with a start and an end and distinguish them from reversals and post reversal turns.– The procedure needs to be explained in greater details with justification of parameter choice.

We have modified the Methods to provide more experimental detail. Videos were scored manually. We measured changes in trajectory (degrees) using the angle measure tool in Fiji as shown in new Figure 3B. We scored forward trajectory changes greater than 50 degrees and reversal-coupled omega turns as reorientations (examples of each in Figure S3). We excluded post-reversal changes in trajectory that did not involve an omega turn. We have clarified this in the Methods. During post hoc analysis of acquired videos, original and new trajectories were set from monitoring movement direction, over sequential frames (~200 frames for forward reorientations and ~600 frames for reversal-coupled omega turns), from the start of the reorientation (original trajectory) to completion of the reorientation (new trajectory) (Figure 3B, S3). Reorientations were scored only in instances where the animal moved a minimum of 3 s (~100 frames @ 30 fps) along a new trajectory.

– How did the authors validate that the procedure performed well, especially during the complex ARS behaviors?

We have clarified the procedure we used for manual scoring by adding examples of sequential frames during reorientation with overlay of the angle measured (Figure 3B, S3) and added representative images showing forward reorientations and reversal coupled turns. We also note that our methods cleanly distinguished ARS (0-5 minutes after removal from food) and dispersal (30-35 minutes off food) behaviors in wild type, demonstrating its effectiveness.

– Again, example trajectories and movies should be shown.

These are now included.

7) All histogram panels lack statistics, e.g. KS test or appropriate alternatives.

We now include appropriate statistical comparisons by KS test.

Reviewer #2:Ramachandran et al. report the discovery of a *C. elegans* GPCR – CKR-1 – that mediates some of the effects of the cholecystokinin-like neuropeptide NLP-12 on posture and foraging behavior. The discovery of this receptor permits further study of this neuropeptide signaling system, which is conserved from worms to vertebrates. Although CKR-1 is expressed in many neurons, the authors show that its function in SMD head-motorneurons is especially important for control of posture and foraging.The manuscript's strengths include: (1) rigorous characterization of receptor-ligand interactions in vitro, using a cell-based assay for GPCR activation, and in vivo, using genetic analysis, (2) compelling data in support of a model in which NLP-12 regulates SMD neurons to control foraging, and (3) high-resolution analysis of *C. elegans* posture during foraging, which illustrates the complexity and richness of this behavior, and (4) the circuit model, i.e. a role for SMDs, is tested using a number of independent methods and clearly indicated.

We thank the reviewer for their positive assessment.

The manuscript does have some weaknesses. In addition to specific technical points listed below, the manuscript discussed neuropeptides derived from a single source, the DVA pre-motor neuron, acting on distinct targets via distinct receptors in a conditional manner. This interesting model is suggested by the title and the abstract and comes up plainly in the introduction and discussion. However, the model is not clearly supported by the data, which primarily focus on the characterization of CKR-1 as a relevant receptor for NLP-12 peptides.

We aim to understand how NLP-12 signaling regulates motor transitions that are characteristic features of both local food searching and oxygen chemotaxis. We demonstrate that NLP-12 can activate 2 GPCRs, CKR-1 and CKR-2. Roles for CKR-2 in regulation of body wall motor neuron activity have been demonstrated previously (Hu et al., 2011), but *ckr-2* deletion has little effect on local searching behavior. We therefore focused our efforts on understanding whether CKR-1 may be a primary target of NLP-12 to promote local searching. Our deletion, rescue, overexpression, cell ablation, photostimulation, silencing and calcium imaging experiments support a model where NLP-12 activates CKR-1 expressed in the SMDs to promote searching. We show that CKR-1 and CKR-2 have largely nonoverlapping expression, suggesting they may be differentially utilized to promote alternate behavioral outcomes. Consistent with this, we are able to assign a functional role for CKR-2 solely during basal locomotion. As prior studies have demonstrated CKR-2 function in body wall motor neurons, we did not pursue this aspect further.

Another weakness in the manuscript arises from the authors' switching between various assays for posture during locomotion, which makes it difficult for the reader to compare data between figures. Rich kymography data are relegated to supplementary figures, and data from only a subset of relevant genotypes are shown as kymographs. The manuscript would be strengthened by more uniform analysis of posture and foraging.

We apologize for not making this more clear in the initial submission. The single worm tracking used for analysis of basal locomotion in Figure 2 affords higher resolution and allows for rich quantification of relatively subtle postural changes. However, we were not able to continuously track animals during the numerous steep turns performed during ARS or with NLP-12 or CKR1 overexpression using this approach (the head and tail were often misassigned during deep bends). Hence, we switched to recording videos of behaving animals, and performing post hoc analysis of bending angles with Wormlab (MBF Biosciences). This approach did not offer the same resolution but proved more robust for analyzing deeper body bends. We measure the body bending angle at the midbody vertex in both analyses. While we made every effort to keep these angle measurements consistent across platforms, we acknowledge the measured angles differ somewhat. We now include discussion of these points in both the Methods and Results. See also response to point 4 below.

Finally, while the data clearly show that effects of NLP-12 on posture and foraging require SMD neurons, the manuscript does not investigate how NLP-12 affects SMD activity. The manuscript would be strengthened by experiments showing a functional connection between DVA and SMD neurons, e.g. functional imaging of SMDs during optogenetic manipulation of DVAs.

We have now added calcium imaging (Figure 7E-F) and SMD silencing (Figure 7D) experiments to address the reviewer’s comments. Details below.

1. One premise of the work is that DVA neurons are the sole source in vivo of NLP-12 peptides. A recent study (Tao et al. 2019, Dev. Cell) shows that there is an alternate source of NLP-12, the PVD nociceptors. The authors should address the possibility that their assays also detect a contribution of PVD neurons to posture/foraging.

We thank the reviewer for this comment. We now include additional data showing that PVD-specific expression of *nlp-12* does not rescue reorientations during ARS (0-5 minutes off food) [Figure S5C]. We also note that recent CeNGEN single cell RNAseq data show high levels of *nlp-12* transcript in DVA and by comparison very low levels in PVD, consistent with our findings.

2. The text associated with Figure 1B-C is tentative with respect to assigning redundant functions to CKR-1 and CKR-2. Why? The data are clear; these receptors function redundantly.

We have modified the text as suggested.

3. The very nice in vitro analysis of NLP-12 receptors should include negative controls. Ideally, the authors would use a scrambled neuropeptide or a related neuropeptide to demonstrate specificity of the interactions between NLP-12 and CKR-1/2.

We now include empty vector negative controls [Figure S2]. Additionally, we note that no other peptides from the synthetic library of roughly 350 peptides significantly activated either CKR-1 or CKR-2.

4. The different 'bending angles' used in Figures 1 and 2 make it difficult to compare data between figures. Also, the schematics used to explain the bending angles have small fonts and are hard to read.

We have expanded the size of the schematics and increased the font size. The single worm tracking used in Figure 2 affords higher resolution and allows for rich quantification of postural changes during basal locomotion. However, we were unable to continuously track animals during the numerous steep turns performed during ARS or with NLP-12 or CKR-1 overexpression using this approach (the head and tail were often misassigned during deep bends). Conversely, we were unable to reliably detect and quantify the comparatively subtle effects of *nlp-12*, *ckr-1* and *ckr-2* deletion during basal locomotion using Wormlab. The major difference between the two analyses is that the single worm tracker divides the animal into 5 segments (head, neck, midbody, hip, tail). The body bend angle was measured at the midbody vertex, between the neck and hip skeleton vertices. The Wormlab bending angles were measured as the supplement of the angle between the head, midbody and tail vertices. We now include expanded discussion of these points in the Methods.

5. Figure 3E shows the results of a nice experiment in which optogenetic activation of NLP-12-expressing cells – presumably DVA – causes reorientations. The authors assert that this effect requires CKR-1 but not CKR-2. The data, however, suggest that CKR-2 might have an effect. The variance of the data does not allow the authors to reject a null hypothesis, but they err in then assuming that this means that CKR-2 plays no role in the phenomenon. This experiment should be repeated to determine whether there is indeed a specific or privileged role for CKR-1 in mediating NLP-12-dependent reorientations.

We modified Figure 3E to display to a scatterplot of the data and modified the text to more clearly describe the variability in *ckr-2* mutant responses to DVA photostimulation. *ckr-1* deletion led to a clear reduction in DVA-stimulated reorientations, leading us to primarily focus on the requirement for CKR-1. The critical importance of CKR-1 for generating reorientations is further supported by our additional behavioral, overexpression and calcium imaging results.

6. Also, Figure 3E should show raw data – don't show proportional changes – and all Figure 3 should be scatter plots allowing the reader to assess the variance of the data.

We modified Figure 3E to show scatter plot, all figures are now scatter plots.

7. The authors show that effects of receptor overexpression are suppressed by loss of NLP-12 peptides. Is there precedent for this kind of genetic interaction in the literature?

We have noted several previous examples in the literature where the effects of receptor overexpression are suppressed by loss of the ligand. For example, exaggerated body bend posture produced FRPR-4 overexpression is suppressed by *flp-13(lf)* (Nelson et al., 2015). Similarly, hyperactive egg-laying produced by overexpression of the SER-1 serotonin receptor is suppressed by loss of serotonin (*tph-1* mutation) (Fernandez et al., 2020).

8. Also, the authors, assert that suppression of effects of CKR-1 overexpression by loss of NLP-12 shows that NLP-12 peptides are the sole ligands for this receptor (page 9, line 17). It is not clear why the authors reach this conclusion.

We have modified the text to indicate NLP-12 is likely to be the major ligand responsible for CKR-1 activation. *nlp-12* deletion suppresses the behavioral effects of *ckr-1* overexpression, indicating that other endogenous peptides cannot effectively substitute for NLP-12 in eliciting the behavioral changes. This conclusion is further supported by our in vitro studies showing that NLP-12 peptides are the only peptides in the library that significantly activate CKR-1.

9. There are some very nice data that are assigned to supplementary figures but might be better placed in main figures. Figure S3A-B shows data that are integral to the authors' model and could be presented in a main figure. Also, the localization of NLP-12::Venus in DVA axons near SMD processes would be appropriate to show in a main figure. It would be ideal to mark SMDs with a red fluor so that NLP-12::Venus colocalization with SMD processes could be assessed.

As requested, we added images to main Figure 5 showing NLP-12::Venus localization in proximity to SMD processes in the nerve ring. For space and clarity, we felt that S3A-B (Figure S6A-B in revised manuscript) should remain supplemental.

10. The kymography data are nice but incomplete. The authors should show kymographs from strains of all relevant genotypes. This would include: (1) ckr-1(oe); nlp-12, (2) nlp-12, ckr-1, and ckr-2 single mutants, and (3) ckr-1; ckr-2 double mutants.

We have added additional kymographs as requested, and moved the kymographs into main figures. Kymographs of movement during ARS are now shown in Figure 3. Kymographs of movement elicited by overexpression are now shown in Figure 6.

11. Page 12, last paragraph indicates that 'low levels' of expression rescue ckr-1 phenotype – how has the expression level been determined? I guess that the authors refer to the amount of DNA used for transgenesis, not a direct measure of transgene expression – this should be reworded.

We have modified the text.

12. The manuscript would be strengthened by experiments that measured the effect of DVA activation on SMD physiology and what contribution NLP-12 signaling makes to any functional connection between these neurons. One potential impact of this work is that it establishes a nice paradigm for new molecular genetic analyses of neuropeptide signaling. Direct observation of the effects of NLP-12 peptides on SMD neuron physiology would further strengthen the authors' conclusions and suggest mechanisms by which CKR-1 regulates cell physiology.

We tried the experiment suggested by the reviewer. We stimulated DVA in hydrogel immobilized transgenic animals expressing DVA::Chrimson::SL2::BFP and SMD::GCaMP6s::SL2::mCherry, but were unable to detect clear SMD calcium transients that were timed with DVA stimulation. The failure to detect a synaptic response may be due to the immobilized preparation or may reflect limited synaptic connectivity between DVA and SMDs–a single synapse between DVA and SMDVL. We envision that NLP-12 activation of the SMDs occurs primarily through volume transmission which would act over a longer time scale and be quite difficult to measure with this approach. To address the reviewer’s comment, we also performed calcium imaging of SMD activity in behaving animals (Figure 7E-F) during ARS. We found that SMD activity is elevated in ARS compared with dispersal. Deletion of *ckr-1* decreases SMD activity during ARS.

Reviewer #3:In this manuscript, Ramachandran and colleagues describe how cholecystokinin-related NLP-12 neuropeptide signalling in *C. elegans* can regulate two different behavioural programmes, area-restricted search (ARS) and basal locomotion, by conditionally engaging different specific receptors that are expressed in different neuronal targets. They thoroughly characterise the CKR-1 receptor which had not been described previously, and place its function in context with that of the previously known NLP-12 receptor CKR-2. The manuscript gives new insight into an interesting and likely conserved mechanisms of how neuromodulatory systems enable adaptive behaviour by coordinating the action of neural circuits even when they are not directly connected. The conclusions drawn appear solid and are justified by the data presented, and the experimental approaches and results are well documented.

We thank the reviewer for their positive assessment.

The main problem with the work is a certain lack of clarity regarding the separation of the roles of the CKR-1 and CKR-2 receptors on basal locomotion/body bending and head bending/reorientations. Overexpression of NLP-12 places animals in a chronic ARS state, as described in a previous publication. Is the NLP-12 overexpression model representative of the increased reorientation in area restricted search, or of control of undulations in basal locomotion, or both? If it is primarily representative of area restricted search, this would mean that CKR-2, similarly to CKR-1, mediates the chronic ARS state induced by NLP-12 overexpression, because in Figure 1B and C its mutation causes a reduction in the phenotype, and deletion of both ckr-1 and ckr-2 causes a stronger reduction.

We used NLP-12 overexpression as a sensitive way to identify potential receptors. After implicating specific receptors through genetic suppression of peptide overexpression, we pursued behavioral studies of deletion mutants to assess specific contributions of CKR-1 versus CKR-2. NLP-12 overexpression produces behavioral effects that are qualitatively similar to ARS, but more severe. However, we are hesitant to assign further significance to interpretation of behaviors arising from peptide overexpression since they likely reflect high, non-physiological levels of NLP-12 peptides.

Also, it is unconvincing that SMD neurons do not express ckr-2 (see S3D); no comparison of ckr-1 and ckr-2 expression levels in SMD is provided and in fact the CeNGEN data of single cell RNAseq of *C. elegans* neurons shows similar expression of both receptors in SMDD (accessible at cengen.shinyapps.io/SCeNGEA). On the other hand, loss of ckr-2 on its own does not cause a significant reduction in ARS (Figure 3A).To clarify this, the authors could measure the reorientation rate in the nlp-12OE ckr-2 mutant strain.

We apologize for not describing our findings more clearly. We noted weaker and more variable expression of *ckr-2* in the SMD neurons. Variable *ckr-2* expression was restricted to the SMDDs, and was not observed in the SMDVs. We noted more consistent expression of *ckr-1* across both SMDDs and SMDVs. We have now clarified this in the revised manuscript. The CeNGEN data do not distinguish SMDV expression. To strengthen our conclusions, we attempted rescue of *ckr-1(lf);ckr-2(lf)* double mutants with wild type *ckr-1* or *ckr-2*. We found that expression using the *ckr-1*, but not *ckr-2*, provided for rescue [Figure S5A]. Additionally, we show expression of wild type *ckr-1* under control of *ckr-1*, but not *ckr-2*, promoter is sufficient for rescue of *ckr-1(lf)* mutants (Figure S5B). We also now show representative tracks for the *nlp-12OE;ckr-1* and *nlp-12OE;ckr-2* mutant strains (Figure 1A), demonstrating that turning remains elevated in these animals.

Given that ckr-1 overexpression as shown in Figures4-6 increases both body bending amplitude and ARS-like high reorientation rate, the authors offer the interesting possibility that SMD may also affect basal locomotion. I would suggest an experiment that clarifies whether SMD also controls body bending in basal locomotion using the single-worm tracking assay shown in Figure 2A with the SMD-specific ckr-1 rescue strains in a ckr-1 mutant background (as used in figure 7). Also they could measure body bending in the existing data on the SMD::Chrimson optogenetics.

To address this point, we measured body bending in response to SMD photostimulation and found that SMD depolarization leads to a modest increase in body bend amplitude [Figure S8E]. We also now include experiments showing that SMD silencing significantly reduces reorientations during ARS [Figure 7D].

[Editors’ note: further revisions were suggested prior to acceptance, as described below.]

The reviewers are very excited about your work and the improvements made in your revision. However, there are some remaining issues that need to be addressed. As outlined below, the concern of reviewers #1-2 with respect to SMD Ca++ imaging. We assume that this would not require new experiments but further in-depth analysis of the recordings, which should be all feasible in a short time frame.Reviewer #1:The authors have fully addressed my review comments to the previous submission.In the present manuscript, the authors provide SMD Ca++ imaging experiment, which require further clarifications:1) SMD activity in crawling worms has been reported previously by several labs, and all studies found consistently activity related to head-bending (Hendricks et al., 2012; Kaplan et al., 2020; Yeon et al., 2018). From the activity profiles shown in Figure 5, it is not possible to evaluate whether these data can be reproduced by the authors. If movie S9 is representative for these recordings, then SMD activity profiles should relate to head-bending. The fluctuations in Figure 5E rather appear as noise to me. I find it essential that the authors annotate behaviour in these recordings (forward crawling, backward crawling, head bending) and analyse the relationship between SMD activity and locomotion. If any discrepancies to the literature remain, this and possible explanations should be discussed.

We thank the reviewer for highlighting this point. We now show heat maps for all recordings and include behavioral annotations immediately below the heat map for each recording (Figure 8A-C). In addition, we now show that reorientation frequency is correlated with average GCaMP fluorescence intensity in the SMDs during local searching (Figure 8D). However, there was not a strong correlation between peak SMD fluorescence and episodes of forward or backward movement, or reorientations. This may be in part attributable to the way we performed our experiments. We measured combined fluorescence of SMDD and SMDV neurons that themselves have distinct patterns of activation. Notably, Kaplan et al. (2020) also found that SMDD and SMDV activity were not strictly correlated with either forward or reverse command states and instead varied according to locomotor state, consistent with our observations. Our work demonstrates that NLP-12 signaling through CKR-1 promotes a state of heightened SMD activity. Based on our observations, we propose that this state of elevated SMD activity is permissive for performing forward reorientations during ARS.

As noted by the reviewer, prior studies of worms immobilized using microfluidic chips (Hendricks 2012, Shen 2016) and freely moving animals (Yeon 2018, Kaplan 2020) have noted anti-phasic activity between SMDD and SMDV neurons and opposing head/neck musculature during head bending (or head casting). Our studies do not directly address this point. We monitored the combined fluorescence of SMD neurons over longer timescales at lower magnification, offering a view of the summed SMD neuronal activity during ARS and dispersal behaviors. The lower magnification used in our studies simplified measurements from animals freely moving over large areas but limited cellular resolution. Nonetheless, to address the reviewer’s question, we attempted to distinguish the fluorescence of SMDD and SMDV neurons in our recordings; however, we did not feel we could with confidence extract this information. We now include additional discussion of these points in the revised version of the manuscript.

2) Despite the concerns above, I find it surprising and interesting that the authors observe different perhaps baseline ratio values in 0-5min vs 30-35min off-food conditions. How this relates to the different behavioral phenotypes, incorporating our knowledge about SMD physiology remains to be discussed in more detail.Hendricks, M., Ha, H., Maffey, N., and Zhang, Y. (2012). Compartmentalized calcium dynamics in a C. elegans interneuron encode head movement. Nature 487, 99-103.Kaplan, H.S., Salazar Thula, O., Khoss, N., and Zimmer, M. (2020). Nested Neuronal Dynamics Orchestrate a Behavioral Hierarchy across Timescales. Neuron 105, 562-576.e569.Yeon, J., Kim, J., Kim, D.-Y., Kim, H., Kim, J., Du, E.J., Kang, K., Lim, H.-H., Moon, D., and Kim, K. (2018). A sensory-motor neuron type mediates proprioceptive coordination of steering in *C. elegans* via two TRPC channels. PLoS biology 16, e2004929.

We now provide additional discussion of physiological regulation of SMD activity and how this is related to our findings. Prior studies have indicated the SMDs are cholinergic, and their stimulation is sufficient to produce ca^2+^ transients in head/neck muscles, consistent with proposed roles in head bending. However, physiological regulation of SMD activity is complex and involves reciprocal connections with RIA interneurons, reciprocal signaling with RME motor neurons, as well as proprioceptive feedback. In particular, inhibitory signaling from the GABAergic RME neurons onto the SMDs is implicated in modulation of head bending amplitude to optimize head bends for forward movement. While the precise role of NLP-12 modulation of SMD activity remains unclear, one intriguing possibility is that NLP12-elicited increases in SMD activity uncouple the SMDs from RME inhibitory regulation, perhaps promoting large amplitude head swings that couple to forward reorientations during searching.

Reviewer #2:The revised manuscript of Ramachandran, Francis and colleagues addresses many of the questions raised during the initial round of review. I have only one comment. The authors include new data reporting SMD activity during area-restricted search and dispersal. This is an interesting experiment that shows clear evidence for CKR-1 in regulation of SMDs. It is not clear why the authors only show data from 3 individuals in panel E of Figure 7; panel F indicates that there are many more individuals that were assayed. I suggest that the authors show data from all the individuals in panel E.

We thank the reviewer for this suggestion. We now include heat maps for all individuals tested in our calcium imaging experiments (Figure 8A-C in revised manuscript).